# Computational design of peptides to target NaV1.7 channel with high potency and selectivity for the treatment of pain

**Phuong T Nguyen[1†], Hai M Nguyen[2†], Karen M Wagner[3], Robert G Stewart[1], Vikrant Singh[2], Parashar Thapa[1], Yi-Je Chen[2], Mark W Lillya[1], Anh Tuan Ton[4], Richard Kondo[4], Andre Ghetti[4], Michael W Pennington[5], Bruce Hammock[3], Theanne N Griffith[1], Jon T Sack[1,6], Heike Wulff[2]\*, Vladimir Yarov-Yarovoy[1,6,7]\***

[1]Department of Physiology and Membrane Biology, University of California Davis, Davis, United States; [2]Department of Pharmacology, University of California Davis, Davis, United States; [3]Department of Entomology and Nematology & Comprehensive Cancer Center, University of California Davis, Davis, United States; [4]AnaBios Corporation, San Diego, United States; [5]Ambiopharm Inc, North Augusta, United States; [6]Department of Anesthesiology and Pain Medicine, University of California Davis, Davis, United States; [7]Biophysics Graduate Group, University of California Davis, Davis, United States

**\*For correspondence:**
hwulff@ucdavis.edu (HW);
yarovoy@ucdavis.edu (VY-Y)

†These authors contributed
equally to this work

**Abstract** The voltage-gated sodium NaV1.7 channel plays a key role as a mediator of action potential propagation in C-fiber nociceptors and is an established molecular target for pain therapy. ProTx-II is a potent and moderately selective peptide toxin from tarantula venom that inhibits human NaV1.7 activation. Here we used available structural and experimental data to guide Rosetta design of potent and selective ProTx-II-based peptide inhibitors of human NaV1.7 channels. Functional testing of designed peptides using electrophysiology identified the PTx2-3127 and PTx2-3258 peptides with $IC_{50}$s of 7 nM and 4 nM for hNaV1.7 and more than 1000-fold selectivity over human NaV1.1, NaV1.3, NaV1.4, NaV1.5, NaV1.8, and NaV1.9 channels. PTx2-3127 inhibits NaV1.7 currents in mouse and human sensory neurons and shows efficacy in rat models of chronic and thermal pain when administered intrathecally. Rationally designed peptide inhibitors of human NaV1.7 channels have transformative potential to define a new class of biologics to treat pain.

## Editor's evaluation

Chronic pain is a major health issue that is notably lacking pharmacological, non-opioid treatment options. The authors of this manuscript have deployed a powerful combination of experimental and computational tools to optimize the binding and selectivity of a series of peptide toxins derived from the toxin Protoxin II. The authors succeed in obtaining a peptide with improved selectivity and affinity for the human sodium channel Nav1.7, which is a major player in pain transmission and generation. This strategy represents a major advance on the road to obtaining specific peptide inhibitors for the treatment of chronic pain.

## Introduction

More than 25 million Americans suffer from chronic pain (*Nahin, 2015*). Chronic pain originates from tissue or nervous system damage and persists longer than three months (*Treede et al., 2015*). The many causes of chronic pain include surgery, chemotherapy, complex regional pain syndrome, and

back pain. People with chronic pain experience higher anxiety, depression, sleep disturbances, and gain weight due to decreased physical activity. Non-opioid treatment options for chronic pain are limited (*Seal et al., 2017*). Inhibitors of neuronal ion channels are important alternatives that have not demonstrated addiction liability. Non-selective $Na_V$ channel inhibitors, including carbamazepine, lacosamide, and lamotrigine are used among initial options to treat patients with chronic pain (*Beyreuther et al., 2007*; *Wiffen et al., 2011*; *Wiffen et al., 2014*). For example, intravenous infusion of the local anesthetic lidocaine, a non-specific $Na_V$ channel inhibitor, reduces chronic pain in some patients (*Hutson et al., 2015*; *Iacob et al., 2018*; *Kandil et al., 2017*; *van der Wal et al., 2016*). However, lidocaine treatments have serious side effects including cardiac arrest, abnormal heartbeat, and seizures. Patients with chronic pain who are not responding to $Na_V$ channel inhibitors can be prescribed opioids, but the severe side effects of opioids such as constipation, respiratory depression, and addiction limit their utility. Intrathecal infusion of the voltage-gated calcium channel inhibitor ziconotide is also effective against chronic pain (*Bäckryd, 2018*; *Deer et al., 2018*) but accompanied by serious psychiatric side effects (*Bäckryd, 2018*). Consequently, the treatment of chronic pain remains a major unmet medical need. $Na_V$ channels have been thoroughly clinically validated as pharmacological targets for pain treatment, but currently available therapies are limited by incomplete efficacy and significant side effects (*Bhattacharya et al., 2009*; *Dib-Hajj et al., 2010*; *Kaczorowski et al., 2008*; *Liu and Wood, 2011*; *Mulroy, 2002*; *Walia et al., 2004*).

Nociceptive signals originate in peripheral nerve fibers that transduce chemical, mechanical, or thermal stimuli into action potentials that propagate along their axons to the synaptic nerve terminals in the spinal dorsal horn (*Basbaum et al., 2009*; *Dib-Hajj et al., 2010*; *Dib-Hajj et al., 2013*; *Waxman and Zamponi, 2014*). Voltage-gated sodium ($Na_V$) channels are key molecular determinants of action potential generation and propagation in excitable cells. Of the nine known human $Na_V$ ($hNa_V$) channel subtypes (*Catterall et al., 2005*), genetic and functional studies identified three subtypes as important for pain signaling: $Na_V1.7$, $Na_V1.8$, and $Na_V1.9$, which are predominantly expressed in peripheral neurons (*Bennett et al., 2019*; *Black et al., 2008*; *Cox et al., 2006*; *Cummins et al., 2004*; *Dib-Hajj et al., 2010*; *Dib-Hajj et al., 2013*; *Estacion et al., 2009*; *Fertleman et al., 2006*; *Goldberg et al., 2012*; *Nassar et al., 2004*; *Reimann et al., 2010*; *Shields et al., 2012*; *Yang et al., 2012*; *Yang et al., 2004*). $Na_V1.7$ possesses a slow closed-state inactivation compared with other channels (*Herzog et al., 2003*), making it uniquely important for setting the threshold for action potential firing, and thus the gain in pain signaling neurons (*Dib-Hajj et al., 2007*; *Rush et al., 2007*). In accordance with this, loss-of-function mutations in $hNa_V1.7$ have been identified in families with congenital insensitivity to pain (*Cox et al., 2006*). Gain-of-function mutations in $hNa_V1.7$ lead to inherited pain disorders; families with inherited erythromelalgia have $hNa_V1.7$ mutations that shift its voltage-dependence of activation to hyperpolarized voltages, leading to hyperexcitability in dorsal root ganglion (DRG) neurons and chronic neuropathic pain (*Cummins et al., 2004*; *Yang et al., 2004*); patients with paroxysmal extreme pain disorder have defects in $hNa_V1.7$ fast inactivation resulting in persistent sodium currents and episodic burning pain (*Fertleman et al., 2006*). These and other studies have validated $hNa_V1.7$ as a prime target for the treatment of pain (*Dib-Hajj et al., 2010*; *Dib-Hajj et al., 2013*; *Waxman and Zamponi, 2014*).

Mammalian $Na_V$ channels are composed of four homologous domains (I through IV), each containing six transmembrane segments (S1 through S6), with segments S1-S4 of the channel forming the voltage-sensing domain (VSD) and segments S5 and S6 forming the pore (*Ahern et al., 2016*; *Payandeh et al., 2011*; *Shen et al., 2019*; *Shen et al., 2017*). The binding of local anesthetics to a receptor site formed within the pore inner cavity can directly block ion conduction through the $Na_V$ channels (*Ragsdale et al., 1994*; *Yarov-Yarovoy et al., 2001*; *Yarov-Yarovoy et al., 2002*). However, because of the high conservation of residues forming this local anesthetic receptor site among the different isoforms, all currently available therapeutic drugs targeting $Na_V$ channels are non-specific.

There is a growing trend in industry and academia to target ion channels with biologics (*Bosmans and Swartz, 2010*; *Neff and Wickenden, 2021*; *Payandeh and Hackos, 2018*; *Wulff et al., 2019*). More than 10 years ago scientists at Merck demonstrated that a peptide from the venom of the Peruvian green velvet tarantula *Thrixopelma pruriens*, termed Protoxin-II (ProTx-II), selectively targeted the $Na_V1.7$ channel subtype and blocked action potential propagation in nociceptors (*Schmalhofer et al., 2008*). Amgen also developed peptide inhibitors of $Na_V1.7$ and identified a novel peptide toxin from the venom of the Chilean tarantula *Grammostola porteria*, termed GpTx-1, which was a less potent

inhibitor of human Na$_V$1.7, compared with ProTx-II, but had 20-fold and 1000-fold selectivity against Na$_V$1.4 (predominantly expressed in muscle) and Na$_V$1.5 (predominantly expressed in the heart) (*Murray et al., 2015*). Using the GpTx-1 NMR structure as a guide, Amgen scientists created a variant with improved potency and selectivity compared with the wild-type toxin, concluding that GpTx-1 variants can potentially be further developed as peptide therapeutics (*Murray et al., 2015*). The most advanced reported preclinical development of Na$_V$-selected peptides is from Janssen Biotech, which demonstrated that ProTx-II exerted a strong analgesic effect following intrathecal injection in rat models of thermal and chemical nociception. While efficacious, ProTx-II had a narrow therapeutic window, and induced profound motor effects at moderately higher doses, consistent with inhibition of Na$_V$ channel subtypes present on motor neurons (Na$_V$1.1 and Na$_V$1.6) (*Flinspach et al., 2017*). Janssen Biotech pursued resource-intensive optimization of ProTx-II, but without a structure to guide optimization. This blind optimization process produced 1500 ProTx-II variants, including a peptide, named JNJ63955918, with at least 100-fold selectivity for Na$_V$1.7 over all other Na$_V$ channel subtypes tested. However, JNJ63955918 had ~10 fold reduced affinity for Na$_V$1.7 (*Flinspach et al., 2017*). The in vivo safety window for JNJ63955918 was 7–16-fold, limited by motor deficits and muscle weakness, consistent with insufficient selectivity against off-target Na$_V$ channels (*Flinspach et al., 2017*). More recently, Merck developed ProTx-II analogues with improved selectivity for Na$_V$1.7, reduced ability to cause mast cell degranulation, and enhanced in vivo profile (*Adams et al., 2022*).

While these prior and ongoing efforts have not succeeded in developing peptides with a sufficiently wide in vivo safety window, the premise that Na$_V$ channel blocking peptide affinity and selectivity could be further optimized remains valid (*Payandeh and Hackos, 2018*). Furthermore, several high-resolution structures of peptide toxins complexes with human Na$_V$ channels were solved recently (*Clairfeuille et al., 2019*; *Pan et al., 2019*; *Shen et al., 2019*; *Xu et al., 2019*), providing essential templates for the structure-guided design of novel therapeutics. These structures revealed key molecular determinants of ProTx-II interaction with the hNa$_V$1.7 channel in both deactivated and activated states (*Shen et al., 2019*; *Xu et al., 2019*). To overcome past issues with peptide optimization, we used the Rosetta computational protein redesign approach, available experimental data, and functional testing of designed peptides using electrophysiological assays, mouse and human sensory neurons, stability assays, and efficacy testing in animal models of pain to generate high-affinity, selective inhibitors of human Na$_V$1.7 channels. Our lead peptides have better potency and selectivity than Janssen's most potent and selective ProTx-II variant. Our lead peptide inhibits sodium current in human and mouse sensory neurons, is stable in artificial cerebrospinal fluid, and is active in rat models of thermal and chronic pain.

## Results
### Design of ProTx-II based peptides targeting hNa$_V$1.7

To optimize potency and selectivity of ProTx-II based peptides to target hNa$_V$1.7, we analyzed x-ray and cryoEM structures of ProTx-II – hNa$_V$1.7 complexes (*Shen et al., 2019*; *Xu et al., 2019*), explored available experimental data on hNa$_V$1.7 interactions with ProTx-II and its homologs (*Moyer et al., 2018*; *Murray et al., 2015*; *Park et al., 2014*; *Wu et al., 2018*; *Xu et al., 2019*; *Zeng et al., 2007*), modeled specific interactions of ProTx-II substitutions with hNa$_V$1.7, and designed new ProTx-II variants using Rosetta (*Bender et al., 2016*; *Kuhlman et al., 2003*). We learned in each optimization round which particular combination of mutations resulted in the most potent and selective ProTx-II redesign. Mutations that improved the potency and selectivity of ProTx-II-based peptides were kept in the following round(s) of optimization. Our interdisciplinary and iterative peptide optimization approach is described below and outlined in *Figure 1*.

### 1st optimization round

During the first round of optimization, we introduced multiple ProTx-II substitutions guided by available experimental data and insights from the cryoEM structure of the ProTx-II – Na$_V$Ab/hNa$_V$1.7 chimera complex in a deactivated state (PDB: 6N4R) (*Xu et al., 2019*). To improve potency in all of our ProTx-II based peptides, we used the C-terminal amidation based on previously published data (*Park et al., 2014*). The ProTx-II – Na$_V$Ab/hNa$_V$1.7 structure revealed that ProTx-II residues W5 and M6 are positioned in the membrane hydrophobic core and make contact with the unique residue

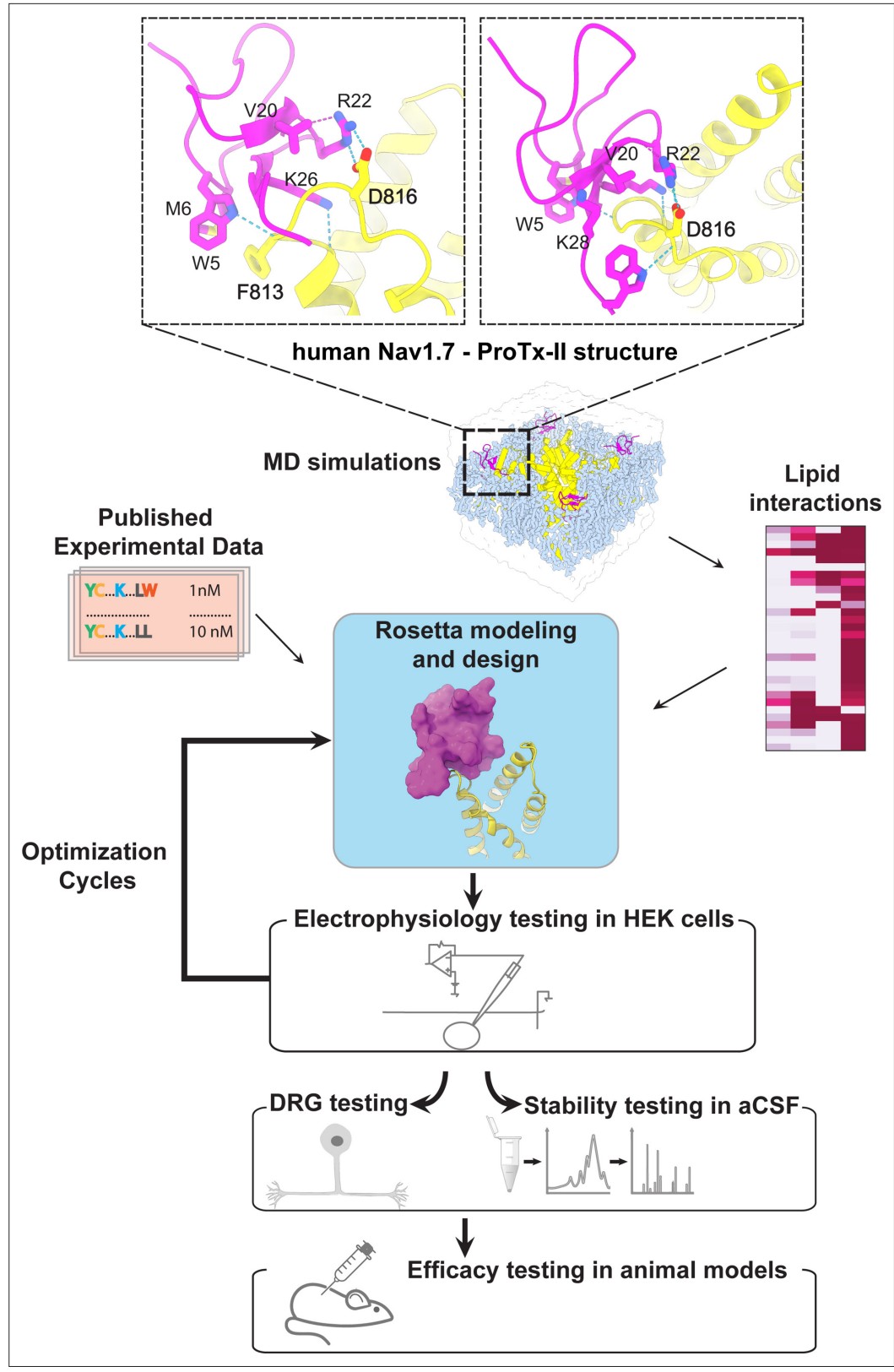

**Figure 1.** The ProTx-II based peptide optimization approach. Top, transmembrane (left) and extracellular (right) views of the wild-type ProTx-II – hNav1.7 structure in a deactivated state (*Xu et al., 2019*) Key residues on the wild-type ProTx-II are shown in stick representation and labeled. Bottom, interdisciplinary peptide optimization approach involving Rosetta design, molecular dynamics (MD) simulations, peptide synthesis and folding,

*Figure 1 continued on next page*

*Figure 1 continued*

electrophysiological testing, peptide stability testing, efficacy in mouse and human DRG neurons, and efficacy in animal models of pain.

The online version of this article includes the following figure supplement(s) for figure 1:

**Figure supplement 1.** Molecular dynamics simulations setup for ProTx-II - human Nav1.7 complex.

**Figure supplement 2.** Multiple sequence alignment of the wild-type ProTx-II and homologous peptide toxins.

---

F813 on the S3 segment of hNa$_V$1.7 VSD-II (F812 in the Na$_V$Ab/hNa$_V$1.7 structure) (*Xu et al., 2019*; *Figure 1* and *Figure 1—figure supplement 1*). We introduced the W5A and M6F substitutions in ProTx-II with the insight from an Amgen's study showing that the double mutant F5A/M6F on GpTx-1 (ProTx-II homolog) improved selectivity for hNa$_V$1.7 over hNa$_V$1.4 (*Murray et al., 2015*) and reasoning that optimized interactions with F813 may improve ProTx-II based peptide selectivity. In addition, the ProTx-II – Na$_V$Ab/hNa$_V$1.7 structure revealed that the hydrophobic residue V20 is positioned in a hydrophilic environment and faces the hNa$_V$1.7 VSD-II S3-S4 loop region (*Figure 1* and *Figure 1—figure supplement 1*). Based on the sequence comparison of ProTx-II to other highly potent peptide toxins targeting the hNa$_V$1.7 VSDII S3-S4 loop region (see *Figure 1—figure supplement 2*), we noticed that ProTx-III (hNa$_V$1.7 IC$_{50}$=11.5 nM) has Lysine and JzTx-V (hNa$_V$1.7 IC$_{50}$=0.6 nM) has Arginine (*Cardoso et al., 2015*; *Moyer et al., 2018*) at the position equivalent to the V20 in ProTx-II. Rosetta modeling of the ProTx-II V20R mutant suggested that arginine could form a salt bridge with D816 on the hNa$_V$1.7 VSD-II S3-S4 loop region (*Figure 2A and B*). Because D816 is only present in the hNa$_V$1.7 and hNa$_V$1.6 subtypes among all human Na$_V$ channels (see *Figure 2—figure supplement 1*), we made the V20R substitution to potentially improve selectivity for hNa$_V$1.7. A Genentech study demonstrated that substituting R22 with nor-arginine (norR) and K26 with arginine improves ProTx-II potency to below IC$_{50}$=0.1 nM for hNa$_V$1.7 (*Xu et al., 2019*). Amgen's study demonstrated that substituting K28 with glutamate improves the selectivity of JzTx-V for hNa$_V$1.7 over Na$_V$1.4 and Na$_V$1.5 (*Moyer et al., 2018*). Based on these data, we substituted ProTx-II R22 with norR, K26 with arginine, and K28 with glutamate (*Figure 2A and B*). We also substituted M19 with leucine to improve peptide stability by preventing methionine-dependent oxidation.

We incorporated these substitutions into two designed ProTx-II variants named PTx2-2954 and PTx2-2955 (*Figure 2A*). Specifically, PTx2-2954 contains the W5A, M6F, M19L, V20R, R22norR, and K28E substitutions and the PTx2-2955 variant contains the W5A, M6F, M19L, V20R, R22norR, K26R, and K28E substitutions (*Figure 2A*). The potency of PTx2-2954 and PTx2-2955 for hNa$_V$1.7 was determined using whole-cell voltage-clamp recordings in HEK 293 cells as described in the Methods. PTx2-2955 inhibited hNa$_V$1.7 currents with an IC$_{50}$ of 185 nM (*Figure 2C and D* and *Table 1*). However, PTx2-2954 had no effect on hNa$_V$1.7 currents at 5 μM (*Figure 2A*). We currently have no explanation for why the PTx2-2954 peptide was not active on hNa$_V$1.7 despite having only an arginine versus lysine difference at position 26. Notably, PTx2-2955 included V20R, K26R, and K28E mutations compared with the wild-type ProTx-II which ultimately benefited the potency and selectivity of our top designs (see ***3rd and 4th optimization rounds*** below). Mutations W5A, M6F, and R22norR did not improve potency and selectivity and were eliminated in the following rounds of optimization. Based on these results, PTx2-2955 peptide was selected as the most potent peptide from the 1st optimization round.

## 2nd optimization round

While the potency of PTx2-2955 was not in the low nanomolar range, the molecular interactions revealed by computational modeling were useful for further rounds of optimization. R26 in PTx2-2955 has extensive contacts with VSD-II and forms a salt bridge with E811 (*Figure 3A and B*). In addition, a hydrogen-bonding network is formed between residues R20, E28 on PTx2-2955 with D816 on VSD-II, a unique residue in hNa$_V$1.7 and hNa$_V$1.6 (*Figure 2—figure supplement 1*). We reasoned that such interactions are important for selectivity and given that the ProTx-II – VSD-II protein-protein interface is highly polar, room for further optimization of the molecular interface of ProTx-II and VSD-II may be limited. We preserved these interactions in this round of optimization and explored substitutions at other positions. Specifically, we designed PTx2-3063 based on PTx2-2955 with an extra substitution E12A which was reported to improve the potency of ProTx-II for hNa$_V$1.7 (*Park et al., 2014*). Notably, in the presence of R26, Norarginine at position 22 does not form a salt bridge with D816 on VSD-II

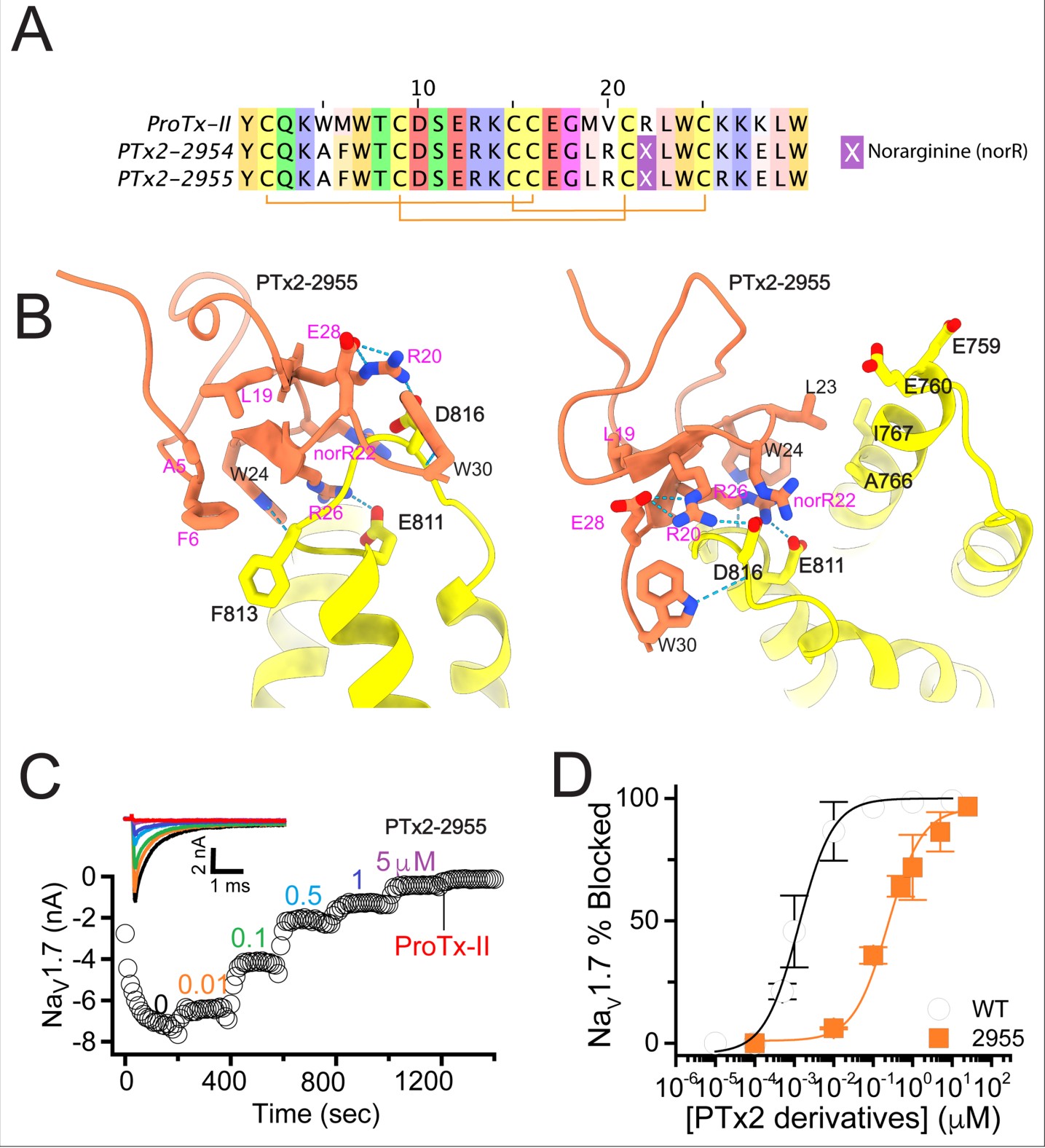

**Figure 2.** The first optimization round. (**A**) Sequence alignment of the wild-type ProTx-II with PTx2-2954 and PTx2-2955 peptides. (**B**) Transmembrane (left panel) and extracellular (right panel) views of the PTx2-2955 – hNa$_V$1.7 model. Key residues on the PTx2-2955 and hNa$_V$1.7 are shown in stick representation and labeled. Nitrogen atoms are colored in blue and oxygen atoms are colored in red. Hydrogen bonds between donor and acceptor atoms are shown by blue dash line. (**C**) Block of whole-cell hNa$_V$1.7 sodium currents by application of increasing concentrations of PTx2-2955 and followed by 1 mM of wild-type ProTx-II as indicated. (**D**) Inhibition of hNa$_V$1.7 currents was measured as shown in C and plotted as a function of WT

*Figure 2 continued on next page*

*Figure 2 continued*

ProTx-2 or PTx2-2955 concentration. Fitting the Hill equation to the data yielded IC$_{50}$ values (95% confidence interval) of 1.7 [0.5, 2.9] nM (n=3) for WT ProTx-II and 185.0 [152.1, 217.9] nM (n=5) for PTx2-2955, respectively.

The online version of this article includes the following figure supplement(s) for figure 2:

**Figure supplement 1.** Sequence alignment of human Nav channel voltage-sensing domain II extracellular regions.

despite being in proximity based on the PTx2-2955 model (*Figure 3B*). We mutated the Norarginine back to Arginine to promote the hydrogen bond with D816 as it appeared in the wt ProTx-II (*Figure 1*) and incorporated this into the design of PTx2-3064. In the presence of R22, the hydrogen bond network at the interacting interface is expanded to E28, R20, and R22 on ProTx-II and D816 on VSDII (*Figure 3B*). We further used Rosetta computational design to explore sequence variants at the non-interface positions of ProTx-II, explicitly looking for substitutions that can stabilize the ProTx-II scaffold or the interface hydrogen bond network while taking into account potential favorable interactions with lipids (see Methods). We also changed the double mutants W5A/M6F back to the wild-type residues in the design process due to the lack of superior engagement with F813 (VSD-II) shown in the PTx2-2955 model. We used Rosetta FastDesign (*Maguire et al., 2021*) to introduce ProTx-II substitutions and design new peptide variants as described in Methods. Among the ProTx-II based peptide consensus sequences designed by Rosetta (*Figure 3—figure supplement 1*), we selected the double mutant S11K/E12D and W7Q to introduce in this round. S11K/E12D allows a salt bridge to be formed between K and D while Q7 forms a hydrogen bond with a backbone carbonyl atom on ProTx-II, thus potentially stabilizing the ProTx-II scaffold and the hydrogen bond network between E28, R20, and R22 on ProTx-II and D816 on VSD-II (*Figure 3B*). We combined these substitutions with other substitutions previously reported to improve potency or selectivity. In particular, the Rosetta suggested substitution W7Q in addition to Y1Q, and W30L was shown to improve selectivity while M19F improved potency for hNa$_V$1.7 (*Flinspach et al., 2015*; *Neff and Wickenden, 2021*). To reduce the potential of misfolding due to multiple substitutions, we strategically introduced these changes into three designed variants PTx2-3065, PTx2-3066, and PTx2-3067.

PTx2-3063 and PTx2-3064 peptides containing the same W5A and M6F mutations as PTx2-2955 inhibited hNa$_V$1.7 currents with IC$_{50}$s of 154 and 52.6 nM, respectively (*Figure 3C and D* and *Table 1*). PTx2-3065, PTx2-3066, and PTx2-3067 peptides containing the wild-type W5 and M6 residues inhibited hNa$_V$1.7 current with IC$_{50}$ values equal to 73.9, 30.8, and 48.3 nM, respectively (*Figure 3D* and *Table 1*). We further tested the selectivity of PTx2-3064 and PTx2-3066 peptides for hNa$_V$1.7 versus other Na$_V$ channels (*Figure 3—figure supplement 2*). PTx2-3064 and PTx2-3066 peptides blocked hNa$_V$1.2 current by ~92 and~41% at 10 μM, respectively. PTx2-3064 and PTx2-3066 peptides blocked hNa$_V$1.5 current by ~25 and~1% at 10 μM, respectively. PTx2-3064 and PTx2-3066 peptides blocked hNa$_V$1.4 current by ~66% and~34% at 10 μM, respectively (*Figure 3—figure supplement 2*). Notably, PTx2-3066 included W7Q, S11K, E12D, and W30L mutations compared with PTx2-2955 which ultimately benefited the potency and selectivity of our top designs (see *3rd and 4th optimization rounds* below). Mutation M19L did not improve potency and selectivity and was eliminated in the following rounds of optimization. Based on these results, PTx2-3066 peptide was selected as the most potent and selective peptide from the 2nd optimization round.

**Table 1.** Potency of redesigned ProTx-II peptides.

| Rank | Peptide | IC$_{50}$ (nM) |
|---|---|---|
| 1 | WT ProTx-II | 0.3–1.7 |
| 2 | PTx2-3258 | 3.8 |
| 3 | PTx2-3128 | 5.0 |
| 4 | PTx2-3127 | 6.9 |
| 5 | PTx2-3361 | 8.6 |
| 6 | Janssen's (JNJ63955918) | 10.0 |
| 7 | PTx2-3260 | 20.8 |
| 8 | PTx2-3066 | 30.8 |
| 9 | PTx2-3259 | 41.8 |
| 10 | PTx2-3067 | 48.3 |
| 11 | PTx2-3064 | 52.6 |
| 12 | PTx2-3065 | 73.9 |
| 13 | PTx2-3063 | 154.0 |
| 14 | PTx2-2955 | 185.0 |
| 15 | PTx2-3126 | 2300.0 |

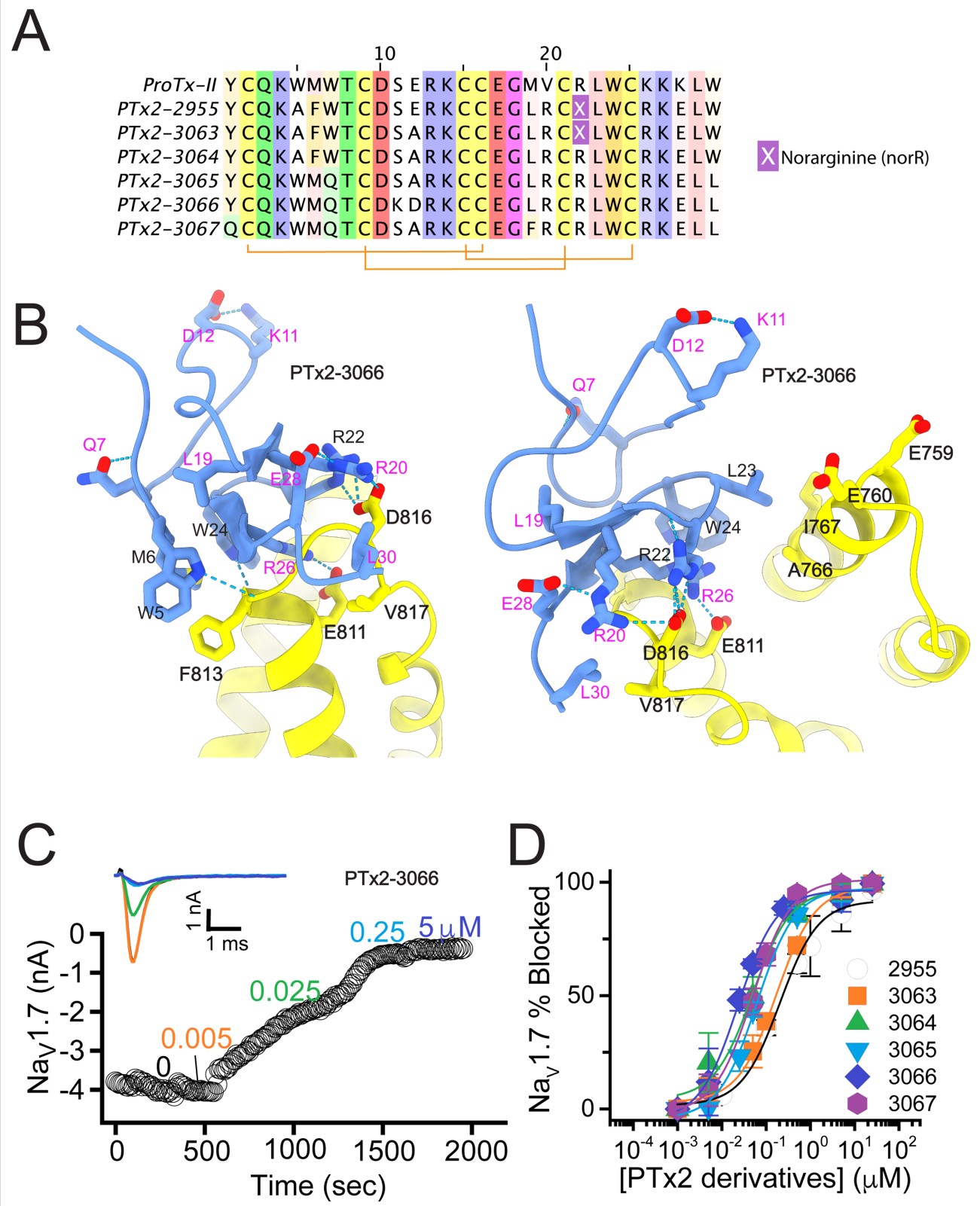

**Figure 3.** The second optimization round. (**A**) Sequence alignment of the wild-type ProTx-II with PTx2-2955 and PTx2-2963 - PTx2-2967 peptides. (**B**) Transmembrane (left panel) and extracellular (right panel) views of the PTx2-3066 – hNa$_V$1.7 model. Key residues on the PTx2-3066 and hNa$_V$1.7 are shown in stick representation and labeled. Nitrogen atoms are colored in blue and oxygen atoms are colored in red. Hydrogen bonds between donor and acceptor atoms are shown by blue dash line. (**C**) Block of whole-cell hNa$_V$1.7 sodium currents by application of increasing concentrations

*Figure 3 continued on next page*

*Figure 3 continued*

of PTx2-3066. (**D**) Inhibition of hNa$_V$1.7 currents was measured as shown in C and plotted as a function concentration of PTx2-2955 or its derivatives. Fitting the Hill equation to the data yielded IC$_{50}$ values (95% confidence interval) of 185.0 [152.1, 217.9] nM (n=5), 154.0 [39.9, 268.1] nM (n=3), nM, 52.6 [7.0, 98.2] nM (n=3), 73.9 [55.8, 92.0] nM (n=4), 30.8 [27.9, 33.7] nM (n=6), and 48.3 [29.5, 67.1] nM (n=4) for PTx2-2955, PTx2-3063, PTx2-3064, PTx2-3065, PTx2-3066, and PTx2-3067, respectively.

The online version of this article includes the following figure supplement(s) for figure 3:

**Figure supplement 1.** Rosetta design of ProTx-II peptides.

**Figure supplement 2.** Selectivity of PTx2-3064 and PTx2-3066.

## 3rd optimization round

Building on the design of PTx2-3066, we explored other combinations of Rosetta suggested substitutions and the reportedly improved potency/selectivity substitutions. Y1Q and M19F from the design of PTx2-3067 were merged into PTx2-3066 with and without the double mutant W5A/M6F to generate new designs PTx2-3126 and PTx2-3127, respectively. In another design, PTx2-3128, we explored whether the scaffold stabilizing double mutant suggested by Rosetta, S11K/E12D, is indeed important for selectivity by introducing the potency improving substitution E12A, which was used in the previous round (***Figure 4A and B***).

The PTx2-3126 peptide containing the W5A and M6F mutations from PTx2-2955 and other mutations from PTx2-3066 inhibited hNa$_V$1.7 currents with an IC$_{50}$=2.3 µM (***Figure 4D*** and ***Table 1***). PTx2-3127 and PTx2-3128 containing the wild-type W5 and M6 residues and other mutations from PTx2-3066 inhibited hNa$_V$1.7 current with IC$_{50}$s equal to 6.9 and 5.0 nM, respectively (***Figure 4D*** and ***Table 1***). We tested the selectivity of PTx2-3127 and PTx2-3128 for hNa$_V$1.7 versus other Na$_V$ channels (see ***Figure 4—figure supplement 1***). PTx2-3127 inhibited other Na$_V$ channels with the following IC$_{50}$ values: 17 µM (hNa$_V$1.1), 5 µM (hNa$_V$1.2), 20 µM (rNa$_V$1.3), 12 µM (hNa$_V$1.4), >137 µM (hNa$_V$1.5), 608 nM (hNa$_V$1.6), >150 µM (hNa$_V$1.8), and 150 µM (hNa$_V$1.9) (see ***Tables 2 and 3***). The data show that PTx2-3127 is at least 1000-fold selective for hNa$_V$1.7 versus hNa$_V$1.1, hNa$_V$1.3, hNa$_V$1.4, hNa$_V$1.5, hNa$_V$1.8, and hNa$_V$1.9. Notably, PTx2-3127 peptide exhibits similar effects on steady-state activation and inactivation on hNav1.7 currents (***Figure 4—figure supplement 2***), suggesting that it retains a similar mechanism of action as ProTx-II and other published ProTx-II derivatives (***Flinspach et al., 2017***; ***Schmalhofer et al., 2008***; ***Smith et al., 2007***; ***Xiao et al., 2010***). However, further improvement is needed for the optimized peptide selectivity for hNa$_V$1.7 versus hNa$_V$1.2 and hNa$_V$1.6. PTx2-3128 inhibited other Na$_V$ channels with the following IC$_{50}$ values: 3.3 µM (hNa$_V$1.1), 570 nM (hNa$_V$1.2), 23 µM (rNa$_V$1.3), 22 µM (hNa$_V$1.4), 34 µM (hNa$_V$1.5), 358 nM (hNa$_V$1.6), 10 µM (hNa$_V$1.8), and 8 µM (hNa$_V$1.9). Notably, PTx2-3127 included M19F mutation compared with PTx2-3066 which ultimately benefited the potency and selectivity of our top designs (see ***4th optimization round*** below). Mutation Y1Q did not improve potency and selectivity and was eliminated in the following round of optimization. Based on these results, PTx2-3127 peptide was selected as the most potent and selective peptide from the 3rd optimization round.

## 4th optimization round

In the final optimization round, we sought to improve the design of PTx2-3127 by introducing substitution Y1H into the design PTx2-3258 (***Figure 5A***). Histidine appeared most frequently in the top Rosetta designs at position 1 (see ***Figure 3—figure supplement 1***). The structural model showed a hydrogen bond formed with a backbone carbonyl atom on ProTx-II (***Figure 5B***) thus potentially stabilizing the ProTx-II scaffold. Building upon PTx2-3258, we replaced Methionine at position 6 by Norleucine to prevent oxidation and incorporated the change in the design of PTx2-3061. All previously tested substitutions selected by Rosetta were hydrogen bond promoting substitutions. In the design of PTx2-3259, we tested if the Q3L substitution suggested by Rosetta (see ***Figure 3—figure supplement 1***) could create an additional stabilizing effect. We selected the third most frequently observed amino acid at this position, Leu based on an experimental design protocol with the membrane scoring function (***Alford et al., 2020***). Lastly, we attempted to explore non-canonical amino acids at positions 27 and 29 to examine whether the selectivity of PTx2-3258 can be improved further given that these positions are near F813 (VSD-II). This resulted in the design of PTx2-3260 with 2,4-dimethyl-phenylalanine and tert-butyl-cysteine at positions 27 and 29, respectively.

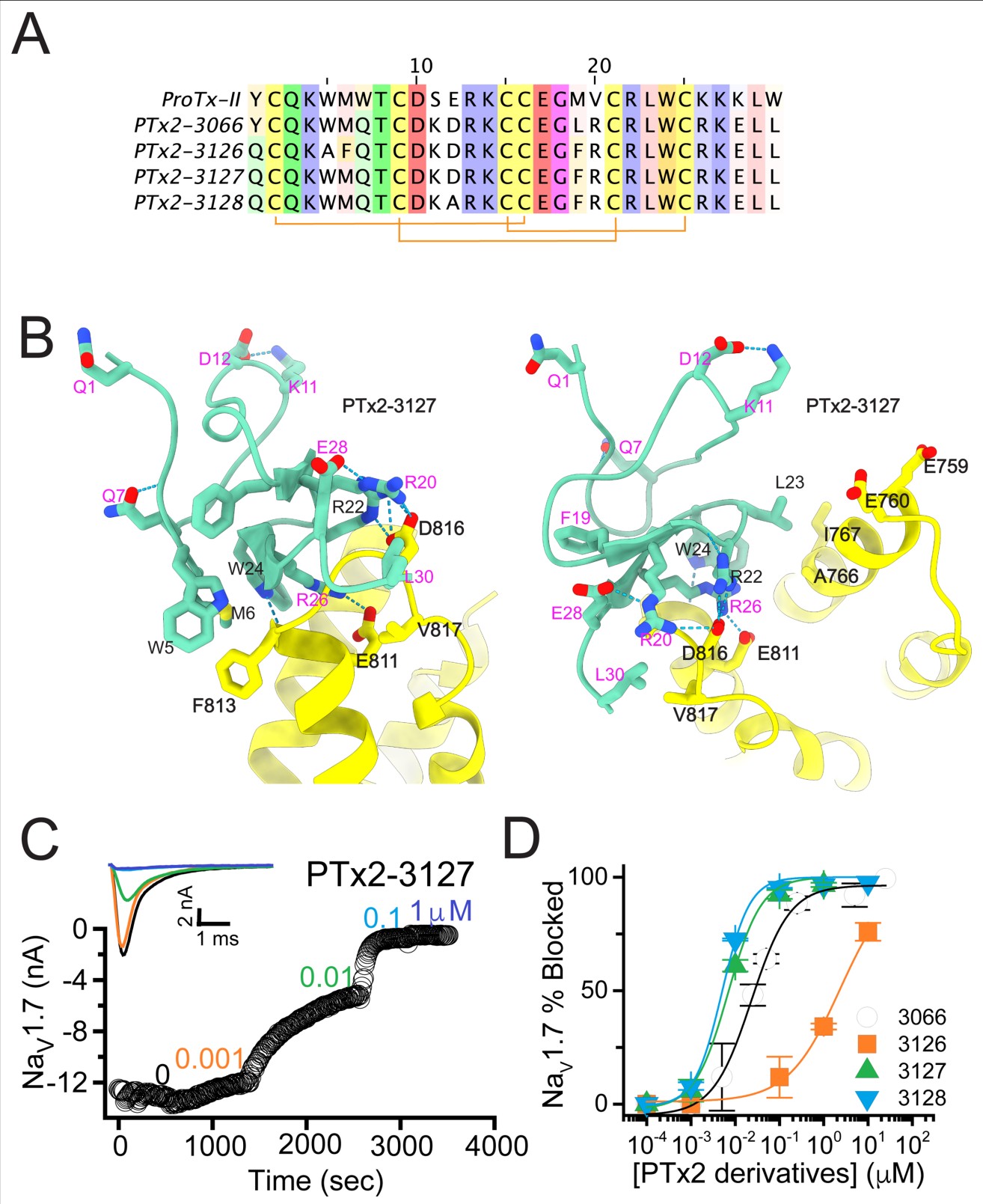

**Figure 4.** The third optimization round. (**A**) Sequence alignment of the wild-type ProTx-II with PTx2-3066 and PTx2-3127 - PTx2-3128 peptides. (**B**) Transmembrane (left panel) and extracellular (right panel) views of the PTx2-3127 – hNaV1.7 model. Key residues on the PTx2-3127 and hNaV1.7 are shown in stick representation and labeled. Nitrogen atoms are colored in blue and oxygen atoms are colored in red. Hydrogen bonds between donor and acceptor atoms are shown by blue dash line. (**C**) Block of whole-cell hNaV1.7 sodium currents by application of increasing concentrations of

*Figure 4 continued on next page*

*Figure 4 continued*

PTx2-3127. (**D**) Inhibition of hNa$_V$1.7 currents was measured as shown in C and plotted as a function concentration of PTx2-3066 or its derivatives. Fitting the Hill equation to the data yielded the IC$_{50}$ values (95% confidence interval) of 30.8 [27.9, 33.7] nM (n=6), 2.3 [1.9, 2.7] µM (n=3), 6.9 [6.7, 7.1] nM (n=3), and 5.0 [4.6, 5.4] nM (n=3) for PTx2-3066, PTx2-3126, PTx2-3127, PTx2-3128, respectively.

The online version of this article includes the following source data and figure supplement(s) for figure 4:

**Figure supplement 1.** Selectivity of PTx2-3127 and PTx2-3128 for hNav1.7 over other sodium channels.

**Figure supplement 1—source data 1.** Mean of individual IC$_{50}$'s (in nM) derived from recordings of 3 or more cells for PTx2-3127 and PTx2-3128 peptides.

**Figure supplement 2.** Mechanism of action of PTx2-3127.

Functional characterization of the activity of the peptides PTx2-3258 – PTx2-3260 and PTx2-3361 on the wild-type hNa$_V$1.7 expressed in HEK 293 cells analyzed by whole-cell voltage-clamp was performed as described in Methods. The ProTx-II variants inhibited the hNa$_V$1.7 channel with the following IC$_{50}$ values: PTx2-3258 (3.8 nM), PTx2-3259 (41.8 nM), PTx2-3260 (21.0 nM), and PTx2-3361 (9.0 nM) (see *Figure 5C and D*, *Figure 5—figure supplement 1*, and *Tables 1–3*). Notably, PTx2-3258 included Y1H mutation compared with PTx2-3127 which ultimately benefited its potency and selectivity.

We tested the broader selectivity of PTx2-3127 and PTx2-3258 on hERG channels. The ProTx-II variants inhibited the hERG channel with the following IC$_{50}$ values: PTx2-3127 (1.9 µM) and PTx2-3258 (1.9 µM) (see *Table 2*). Therefore, PTx2-3127 has 272-fold and PTx2-3258 has 496-fold selectivity for hNa$_V$1.7 versus hERG. Notably, while the wild-type ProTx-II did not inhibit K$_V$2.1 channel at 100–300 nM (*Bosmans et al., 2008*; *Bosmans et al., 2011*; *Schmalhofer et al., 2009*), it inhibited Cav3 channels in the micromolar range (*Bladen et al., 2014*; *Middleton et al., 2002*). We hypothesize that our lead peptides (PTx2-3127 and PTx2-3258) might also inhibit Cav3 channels in the micromolar range and further optimization of peptide selectivity and potency will be needed.

## Stability of designed peptides in artificial cerebrospinal fluid

To access the biologically relevant stability of the wild-type ProTx-II, PTx2-3127, and PTx2-3258, peptides were incubated in artificial cerebrospinal fluid (aCSF) as described in Methods and their stability was determined by HPLC. Notably, the wild-type ProTx-II, PTx2-3127, and PTx2-3258 were found to be stable in aCSF at 37°C for more than 50 hr (*Figure 5—figure supplement 2*).

**Table 2.** Selectivity profile of PTx2-3127 and PTx2-3258 peptides for hNav1.7 versus all other human Nav channels.

| Nav subtype | PTx2-3258 | | PTx2-3127 | |
| --- | --- | --- | --- | --- |
| | IC$_{50}$ (nM) | Selectivity for hNav1.7 vs hNav1.x (fold) | IC$_{50}$ (nM) | Selectivity for hNav1.7 vs hNav1.x (fold) |
| hNav1.1 | 5013 | 1319 | 16,970 | 2459 |
| hNav1.2 | 3399 | 894 | 5040 | 730 |
| rNav1.3 | 14,093 | 3708 | 20,040 | 2904 |
| hNav1.4 | 8877 | 2336 | 11,530 | 1671 |
| hNav1.5 | 38,315 | 10,082 | 137,090 | 19,868 |
| hNav1.6 | 382 | 100 | 608 | 88 |
| hNav1.7 | 3.8 | 1 | 6.9 | 1 |
| hNav1.8 | 43,079 | 11,336 | >150,000 | >20,000 |
| hNav1.9 | 59,443 | 15,642 | >150,000 | >20,000 |
| hERG | 1861 | 496 | 1889 | 272 |

**Table 3.** Comparison of selectivity profiles of PTx2-3127 and PTx2-3258 peptides for hNav1.7 versus hNav1.2, hNav1.4, hNav1.5, and hNav1.6 channels.

| Rank | Peptide | Affinity (IC50) for hNav1.7 (nM) | Selectivity for hNav1.7 vs hNav1.2 (fold) | Selectivity for hNav1.7 vs hNav1.4 (fold) | Selectivity for hNav1.7 vs hNav1.5 (fold) | Selectivity for hNav1.7 vs hNav1.6 (fold) |
|---|---|---|---|---|---|---|
| 1 | PTx2-3258 | 3.8 | 894 | 2336 | 10,082 | 100 |
| 2 | PTx2-3127 | 6.9 | 730 | 1671 | 19,868 | 88 |
| 3 | PTx2-3128 | 5.0 | 114 | 4,500 | 6,800 | 70 |
| 4 | Janssen's (JNJ63955918) | 10 | 160 | 500 | >1000 | 100 |
| 5 | Wild-type ProTx-II | 0.3–1 | 100–140 | 260–380 | 300–1000 | 86 |

## Efficacy of designed peptides on mouse nociceptor DRG neurons

Na$_V$1.7 is important for pain signaling in mice (*Gingras et al., 2014*; *Nassar et al., 2004*). As mice are valuable preclinical models for therapeutic development it is important to know whether mouse endogenous Na$_V$1.7 is responsive to any therapeutic candidate (*Beckley et al., 2021*; *Shiers et al., 2020*). We studied the effects of PTx2-3127 on Na$_V$ currents of genetically identified mouse nociceptor sensory neurons. Mrgprd$^+$ nonpeptidergic nociceptors were identified by fluorescence in *Mrgprd$^{GFP}$* mice (*Zylka et al., 2005*). *Mrgprd$^{GFP}$* DRG neurons from adult mice have significant expression of mRNA for Na$_V$1.7, Na$_V$1.8 and Na$_V$1.9 with other Na$_V$ transcripts in much lower abundance (Na$_V$1.8~Na$_V$1.9>Na$_V$1.7>>Na$_V$1.6>>Na$_V$1.1) (*Zheng et al., 2019*). Presence of Na$_V$1.7 protein in DRG neurons of the *Mrgprd$^{GFP}$* mouse line used for electrophysiology was confirmed by observation of anti-Na$_V$1.7 immunofluorescence in *Mrgprd$^{GFP}$* DRG neuron cell bodies and axonal processes (*Figure 6A*), consistent with prior reports of Na$_V$1.7 localization to small, unmyelinated neurons (*Black et al., 2012*). Anti-Na$_V$1.7 immunofluorescence was variable in *Mrgprd$^{GFP}$* DRG neurons with some exhibiting high and others low density of Na$_V$1.7 protein (*Figure 6A*, DRG inset arrows and arrowhead, respectively).

Application of 1 µM PTx2-3127 to dissociated *Mrgprd$^{GFP}$* neurons under voltage clamp resulted in elimination of a fast-inactivating Na$_V$ component (*Figure 6B*, black trace). Blinded, interleaved experiments with either 1 µM PTx2-3127 or vehicle revealed that PTx2-3127 inhibited 48±17 pA/pF (mean ± SEM) of inward current 0.4–1.0ms after neurons were stepped from –80–0 mV, while vehicle had little effect, inhibiting 2±6 pA/pF (*Figure 6C*, left). Subsequent application of 1 µM TTX to the vehicle controls inhibited 37±12 pA/pF of inward current, similar to the density inhibited by PTx2-3127. Subsequent application of 1 µM TTX to PTx2-3127 had little effect, 2.1±2.8 pA/pF, showing PTx2-3127 inhibits TTX-sensitive currents in *Mrgprd$^{GFP}$* neurons (*Figure 6C*, middle). The density of inhibitor-resistant peak current was similar for TTX ±PTx2-3127 (*Figure 6C*, right). Comparison of Na$_V$ current peak times substantiated the observation that PTx2-3127-sensitive currents were faster than PTx2-3127-resistant currents (*Figure 6D*). In vehicle controls TTX-sensitive peak currents were faster than TTX-resistant peak currents, consistent with a prior study of *Mrgprd$^{GFP}$* neurons (*Dussor et al., 2008*). Overall, the similar effects of either PTx2-3127 or TTX on Na$_V$ currents suggests PTx2-3127 targets the TTX-sensitive channels of *Mrgprd$^{GFP}$* neurons. As *Mrgprd$^{GFP}$* neurons express Na$_V$1.7, which is TTX-sensitive (*Klugbauer et al., 1995*), and have much lower transcript abundances of the other TTX-sensitive channels, Na$_V$1.1, 1.2, 1.3, 1.4, 1.6 (*Zheng et al., 2019*), these results are consistent with PTx2-3127 inhibiting Na$_V$1.7 channels in mouse *Mrgprd$^+$* nonpeptidergic nociceptors.

The impact of the designed peptide on action potential firing of dissociated *Mrgprd$^{GFP}$* neurons was assessed with current-clamp recording. Action potentials were recorded in vehicle, then 1 µM PTx2-3127, then 1 µM TTX. Blinded interleaved controls were conducted with vehicle replacing PTx2-3127. When stimulated with 20ms current injections at 150% of rheobase, the step current required to evoke a single action potential, at 1, 3, and 10 Hz, PTx2-3127 suppressed repetitive firing of most neurons (*Figure 6E*). In 27% of neurons (7 of 24), no block of action potentials was observed even in TTX (*Figure 6—figure supplement 1*), and these were not included in further analyses. Injecting current into DRG neurons to lower resting potential can relieve Nav1.7 inactivation and enhance the reliance of action potential generation on Nav1.7 conductance (*Shields et al., 2018*). However, even

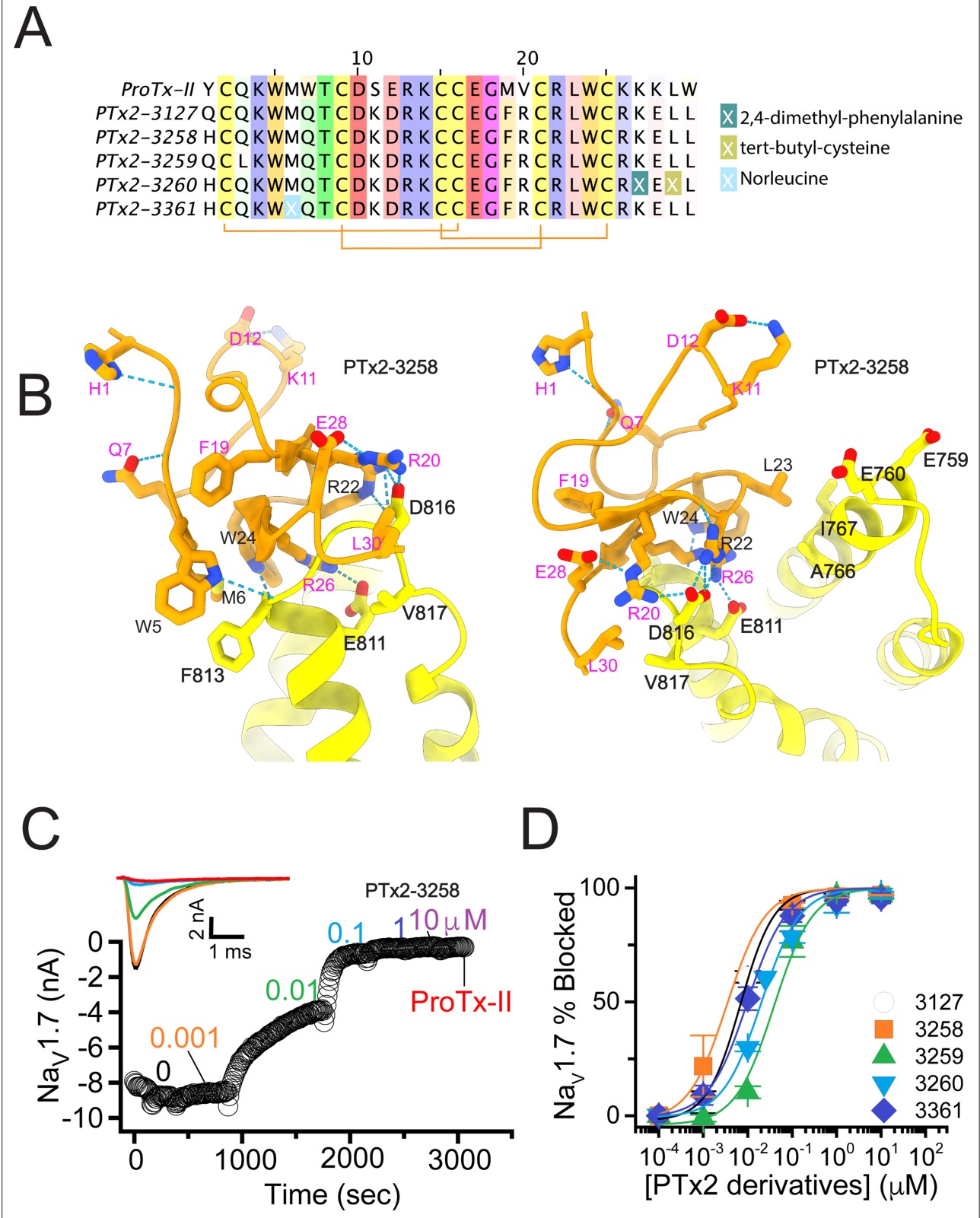

**Figure 5.** The fourth optimization round. (**A**) Sequence alignment of the wild-type ProTx-II with PTx2-3127, PTx2-3258, PTx2-3259, PTx2-3260, and PTx2-3361 peptides. (**B**) Transmembrane (left panel) and extracellular (right panel) views of the PTx2-3258 – hNaᵥ1.7 model. Key residues on the PTx2-3258 and hNaᵥ1.7 are shown in stick representation and labeled. Nitrogen atoms are colored in blue and oxygen atoms are colored in red. Hydrogen bonds between donor and acceptor atoms are shown by blue dash line. (**C**) Block of whole-cell hNaᵥ1.7 sodium currents by application of

*Figure 5 continued on next page*

*Figure 5 continued*

increasing concentrations of PTx2-3258 and followed by 1 mM of wild-type ProTx-II as indicated. (**D**) Inhibition of hNa$_V$1.7 currents was measured as shown in C and plotted as a function concentration of PTx2-3127 or its derivatives. Fitting the Hill equation to the data yielded the IC$_{50}$ values (95% confidence interval) of 6.9 [6.7, 7.1] nM (n=3), 3.8 [0.3, 7.3] nM (n=5), 41.8 [16.5, 67.1] nM (n=3), 20.8 [12.4, 29.2] nM (n=3), 8.6 [5.6, 11.6] nM (n=3), nM for PTx2-3127, PTx2-3258, PTx2-3259, PTx2-3260, and PTx2-3361, respectively.

The online version of this article includes the following source data and figure supplement(s) for figure 5:

**Figure supplement 1.** Selectivity of PTx2-3258 for hNav1.7 over other sodium channels.

**Figure supplement 1—source data 1.** Mean of individual IC$_{50}$'s (in nM) derived from recordings of 3 or more cells for PTx2-3258 peptide.

**Figure supplement 2.** Stability of the wild-type ProTx-II, PTx2-3127, and PTx2-3258 in Artificial Cerebrospinal Fluid.

when currents were injected to hold *Mrgprd*$^{GFP}$ neurons at less than –80 mV (after liquid junction potential correction), we saw similar results with PTx2-3127 (*Figure 6—figure supplement 2*), and in 25% of neurons (4 of 16) no block of action potentials was observed in TTX. In all TTX-sensitive neurons, action potentials were blocked by PTx2-3127, and subsequent application of TTX had little additional effect (*Figure 6F*). Rheobase was also increased by PTx2-3127 (*Figure 6G*). These data demonstrate that PTx2-3127 can inhibit mouse nociceptor excitability.

## Efficacy of designed peptides on human DRG neurons

We studied the effects of PTx2-3127 on the inhibition of single and multiple action potentials properties generated in adult human DRG neurons isolated from a human organ donor. The DRG neurons in culture were treated for 24 hr with 50 μM oxaliplatin to model chemotherapy-induced neuropathy (*Braden et al., 2022*; *Chang et al., 2018*; *Li et al., 2018*). We chose this model as it has been previously shown that pharmacological targeting of Na$_V$1.7 reduces neuropathic pain in this model (*Chang et al., 2018*; *Dustrude et al., 2016*; *Li et al., 2018*). We found that rheobase increased with increasing concentrations of PTx2-3127 (*Figure 7A* and *Table 4*). We then measured action potentials induced by a train of 10–120 individual current steps delivered at 0.1, 1, 3, and 10 Hz, using current injection at 150% of baseline rheobase. The percentage of action potentials remaining was calculated as the number of action potentials in the presence of PTx2-3127 divided by the number of action potentials obtained under control conditions (without drug) at the same frequency. The number of remaining action potentials was reduced in a dose-dependent manner at 0.01, 0.1, and 1 μM PTx2-3127 at different frequencies following 24 hr of incubation with Oxaliplatin (*Figure 7B* and *Table 4*). Our data demonstrate that PTx2-3127 is effective at reducing excitability and action potentials firing in human sensory neurons in an in vitro model of chemotherapy-induced neuropathy.

## Efficacy of designed peptides in animal models of pain

To study the efficacy of PTx2-3127 in animal models of pain, we tested it initially in naïve female and male rats to assess the thermal nociceptive responses and monitor open field activity. Whereas mouse sensory neurons are a useful model in vitro due to their genetic tractability, rats have been proposed to provide more reliable behavioral responses in pain models (*Mogil, 2009*). Doses were selected referencing the in vivo data available for ProTx-II. Merck's study found that ProTx-II had a laming effect via intrathecal administration at 2.4 μg but no effect on nociceptive assays at 0.24 μg i.t. (*Schmalhofer et al., 2008*). Janssen's study reported that 2 μg of ProTx-II in 10 μL was the maximum tolerated dose in rats (*Flinspach et al., 2017*). Based on this information we chose a conservative dose of 1.2 and 1.6 μg in 10 μL for intrathecal administration to naïve rats. Intrathecal administration was performed via implanted cannulas which were surgically placed in the subarachnoid space of the spinal cord between L4 and L5. After recovery from surgery (~7 days) the rats were assessed for gait and mobility prior to peptide dosing.

The 1.6 vs 1.2 μg dose resulted in robust analgesia with several rats reaching the cutoff latency (30 s) for a number of hours on a 52.1°C hotplate assessed once per hour (*Figure 8A and B*) (Two Way Repeated Measures ANOVA, Holm-Sidak method post hoc, p<0.001 PTx2-3127 n=11 vs vehicle n=9). Importantly, this dose did not lame or significantly alter motor activity of the rats. Rats that timed out per the cutoff were immediately ambulatory after being removed from the hot plate. The same dose was administered to a group of rats with oxaliplatin induced neuropathy (*Figure 8C*). These rats with induced chronic pain were assessed on the 52.1°C hotplate to compare to results from naïve rats.

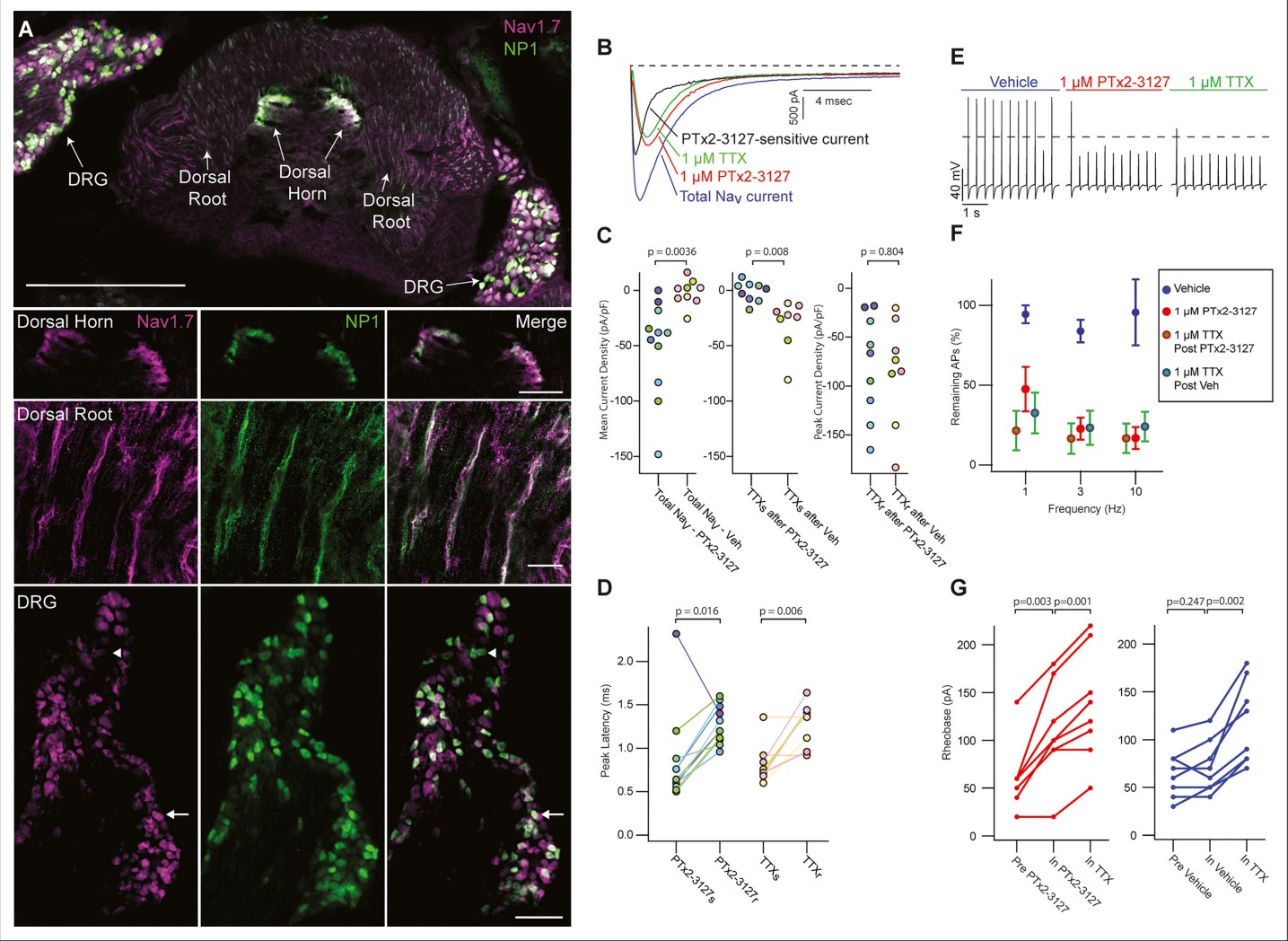

**Figure 6.** Efficacy of designed Na$_V$1.7-selective inhibitor (PTx2-3127) on Na$_V$ channels of mouse nonpeptidergic nociceptor neurons. (**A**) Immunofluorescence from *Mrgprd*$^{GFP}$ labeled NP1 nociceptors (AB_300798, green) and Na$_V$1.7 (AB_2877500, magenta) in a mouse L5 spinal section. Orientation of left DRG was moved during sectioning. Lower panels are zoomed in images to highlight colocalization (white) in dorsal horn nociceptor terminals, dorsal root fibers and DRG cell bodies. NP1 nociceptor DRG cell bodies show both high (arrow) and low (arrowhead) immunofluorescence for Na$_V$1.7. Top image, dorsal horn and DRG zoom images are a z-projection of 3 confocal images spanning 10.06 μm. Zoom in image of dorsal root fibers is a z-projection of 9 airyscan images spanning 3.18 μm. Scale bar in the top image is 500 μm. Scale bars in the dorsal horn, dorsal root and DRG zoom in panels are 100, 20 and 100 μm, respectively. (**B**) Voltage clamp recordings of Na$_V$ currents from dissociated NP1 nociceptors showing impact of PTx2-3127 (red) and subsequent application of TTX (green). Fast-inactivating Na$_V$ component revealed by subtraction of 1 μM PTx2-3127 trace from total Na$_V$ current. Black dotted line represents 0 pA of current. (**C**) Left: Mean current density from 0.4 to 1ms of PTx2-3127 sensitive current and vehicle sensitive current. Middle: Mean current density from 0.4 to 1ms of TTX sensitive current after application of PTx2-3127 or vehicle. Right: Peak current density of TTX resistant current after application of PTx2-3127 or vehicle and TTX. Point colors represent the same neuron (N=4 mice). p values calculated by Students T-Test. (**D**) Peak time of PTx2-3127 sensitive and resistant currents as well as peak time of TTX sensitive and resistant currents. Point colors correspond to the same neurons and is consistent with points shown in C. p values calculated by Students T-Test. (**E**) Current clamp recording of NP1 action potentials and failures with 3 Hz stimuli in vehicle, 1 μM PTx2-3127 and 1 μM TTX. Dashed line represents 0 mV. (**F**) Average remaining NP1 action potentials (APs) versus frequency in PTx2-3127 (red points, n=8 neurons, N=4 mice) or in vehicle control (blue points, n=8 neurons, N=4 mice). Average remaining APs after PTx2-3127 or vehicle control in 1 μM TTX (red circle green fill and blue circle green fill, respectively). Neurons with no sensitivity to TTX were excluded from this analysis. (**G**) Rheobase of NP1 neurons before PTx2-3127, in PTx2-3127 and in TTX (left). Rheobase of NP1 neurons before vehicle, in vehicle and in TTX (right). p values calculated by Students T-test.

The online version of this article includes the following figure supplement(s) for figure 6:

**Figure supplement 1.** Lack of efficacy of PTx2-3127 or TTX on a subset of *Mrgprd*$^{GFP}$ mouse neurons.

**Figure supplement 2.** Effects of PTx2-3127 and TTX on *Mrgprd*$^{GFP}$ mouse neurons held below –80 mV.

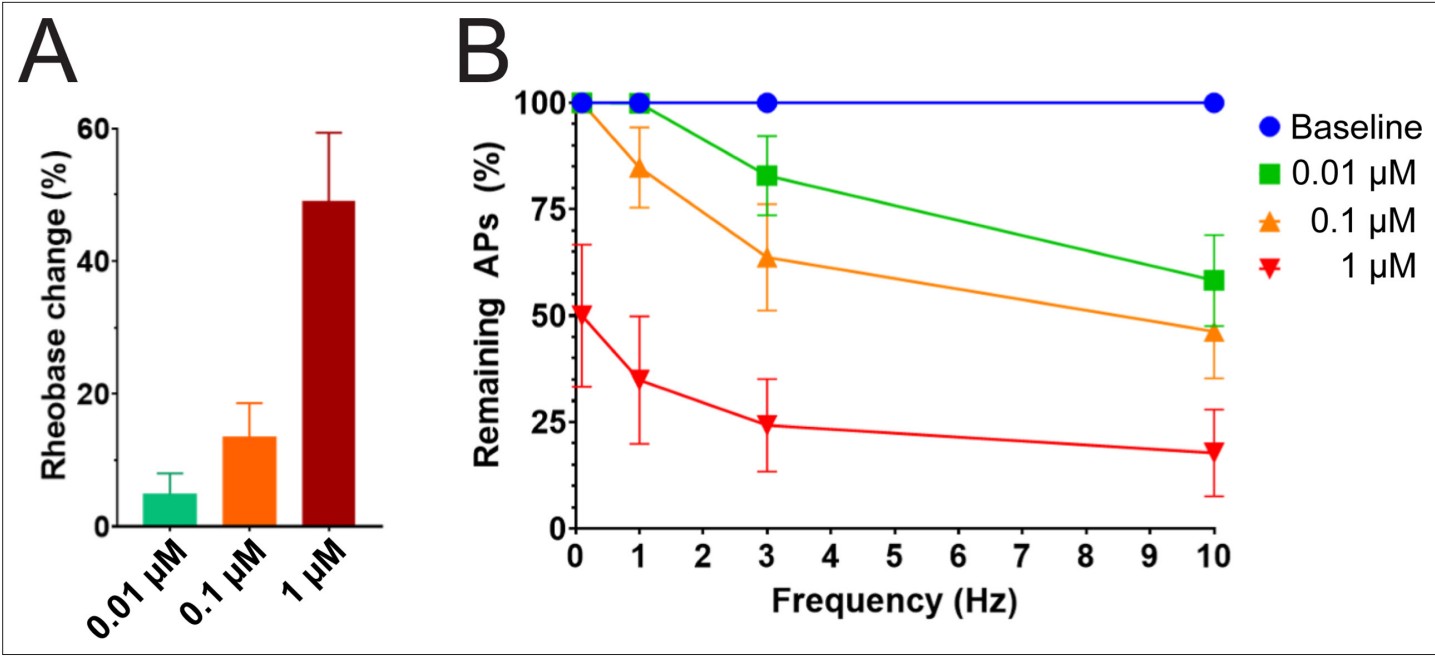

**Figure 7.** Efficacy of PTx2-3127 on rheobase and action potentials in human DRG neurons. (**A**) Efficacy of PTx2-3127 on rheobase in human DRG neurons following 24 h incubation with Oxaliplatin (50 µM). Rheobase after perfusion of the compound is normalized to baseline. (**B**) Efficacy of PTx2-3127 on action potentials (APs) in human DRG neurons following 24 h incubation with Oxaliplatin (50 µM). Action potential inhibition after perfusion of the compound is normalized to baseline. APs were elicited at 150% of baseline rheobase. Results are presented as mean ± SEM.

Again the 1.6 µg i.t. dose of PTx2-3127 resulted in robust analgesia, however, with a slightly different time course of action (Two Way Repeated Measures ANOVA, Holm-Sidak method post hoc, p=0.029 PTx2-3127 n=5 vs vehicle n=4).

## Discussion

Our study provides valuable insights into the development of natural peptide-based therapeutics to treat chronic pain. First, natural peptides, such as ProTx-II, constitute useful starting protein scaffolds for further optimization of selectivity, potency, stability, and bioavailability. Second, high-resolution structures of natural peptide – protein receptor complexes, such as the ProTx-II – hNa$_V$1.7-Na$_V$Ab chimera (*Xu et al., 2019*), and available experimental data on peptide – protein receptor interactions, such as studies by Amgen, Genentech, and Janssen in the case of ProTx-II (*Flinspach et al., 2017*; *Moyer et al., 2018*; *Xu et al., 2019*), provide essential data for the rational design of peptide-based therapeutics. Third, computational structural biology-based protein design using Rosetta allows rational exploration of peptide substitutions in silico guided by high-resolution structures of peptide – protein receptor complexes (*Bender et al., 2016*; *Leman et al., 2020*).

The previous state-of-the-art ProTx-II based peptide optimization by Janssen identified a peptide variant (named JNJ63955018) that achieved in vitro selectivity for hNa$_V$1.7 versus other human Na$_V$ channels ranging from 100-fold (vs hNa$_V$1.6) to more than 1,000-fold (vs hNa$_V$1.5) (*Flinspach et al., 2017*). However, the in vivo safety window for JNJ63955018 peptide was only 7–16 fold (*Flinspach et al., 2017*). Therefore, further improvement of in vitro selectivity for hNa$_V$1.7 versus other human Na$_V$ channels to achieve >1000 fold is necessary to expand the in vivo safety window to at least 100-fold (*Schmalhofer et al., 2008*). Our structure-guided peptide optimization facilitated efficient identification of promising combinations of substitutions and tested only dozens of top candidates compared with the previous comprehensive mutagenesis efforts which synthesized and screened up to 1500 peptide variants (*Flinspach et al., 2015*; *Flinspach et al., 2017*; *Neff and Wickenden, 2021*). Redesign of ProTx-II peptide using Rosetta identified novel and confirmed previously reported substitutions that improved selectivity for hNa$_V$1.7 versus other human Na$_V$ channels while preserving low nanomolar potency (see *Table 5*). Rosetta introduced substitutions of ProTx-II residues facing

**Table 4.** Rheobase and number of action potentials following perfusion of PTx2-3127 following 24 h incubation with Oxaliplatin.

| | [Drug] μM | Rheobase (pA) | Number of APs | | | | % change Rheobase | Remaining AP (%) | | | |
|---|---|---|---|---|---|---|---|---|---|---|---|
| | | | 0.1 Hz | 1 Hz | 3 Hz | 10 Hz | | 0.1 Hz | 1 Hz | 3 Hz | 10 Hz |
| | Baseline | 500 | 10 | 120 | 120 | 62 | 0.0 | 100.0 | 100.0 | 100.0 | 100.0 |
| | 0.01 | 480 | 10 | 120 | 120 | 61 | –4.0 | 100.0 | 100.0 | 100.0 | 98.4 |
| | 0.1 | 460 | 10 | 120 | 120 | 60 | –8.0 | 100.0 | 100.0 | 100.0 | 96.8 |
| Cell 1 | 1 | 640 | 10 | 120 | 120 | 59 | 28.0 | 100.0 | 100.0 | 100.0 | 95.2 |
| | Baseline | 300 | 10 | 120 | 120 | 120 | 0.0 | 100.0 | 100.0 | 100.0 | 100.0 |
| | 0.01 | 280 | 10 | 120 | 120 | 47 | –6.7 | 100.0 | 100.0 | 100.0 | 39.2 |
| | 0.1 | 320 | 10 | 120 | 120 | 24 | 6.7 | 100.0 | 100.0 | 100.0 | 20.0 |
| Cell 2 | 1 | 600 | 0 | 0 | 0 | 0 | 100.0 | 0.0 | 0.0 | 0.0 | 0.0 |
| | Baseline | 360 | 10 | 120 | 120 | 37 | 0.0 | 100.0 | 100.0 | 100.0 | 100.0 |
| | 0.01 | 460 | 10 | 120 | 96 | 28 | 27.8 | 100.0 | 100.0 | 80.0 | 75.7 |
| | 0.1 | 520 | 10 | 53 | 36 | 16 | 44.4 | 100.0 | 44.2 | 30.0 | 43.2 |
| Cell 3 | 1 | 700 | 0 | 0 | 0 | 0 | 94.4 | 0.0 | 0.0 | 0.0 | 0.0 |
| | Baseline | 1450 | 10 | 120 | 120 | 32 | 0.0 | 100.0 | 100.0 | 100.0 | 100.0 |
| | 0.01 | 1550 | 10 | 119 | 8 | 1 | 6.9 | 100.0 | 99.2 | 6.7 | 3.1 |
| | 0.1 | 1650 | 10 | 105 | 1 | 1 | 13.8 | 100.0 | 87.5 | 0.8 | 3.1 |
| Cell 4 | 1 | 1800 | 10 | 0 | 0 | 0 | 24.1 | 100.0 | 0.0 | 0.0 | 0.0 |
| | Baseline | 1800 | 10 | 120 | 120 | 48 | 0.0 | 100.0 | 100.0 | 100.0 | 100.0 |
| | 0.01 | 2000 | 10 | 120 | 105 | 28 | 11.1 | 100.0 | 100.0 | 87.5 | 58.3 |
| | 0.1 | 2000 | 10 | 120 | 90 | 27 | 11.1 | 100.0 | 100.0 | 75.0 | 56.3 |
| Cell 5 | 1 | 2100 | 10 | 120 | 30 | 3 | 16.7 | 100.0 | 100.0 | 25.0 | 6.3 |
| | Baseline | 400 | 10 | 120 | 120 | 52 | 0.0 | 100.0 | 100.0 | 100.0 | 100.0 |
| | 0.01 | 420 | 10 | 120 | 75 | 7 | 5.0 | 100.0 | 100.0 | 62.5 | 13.5 |
| | 0.1 | 520 | 10 | 20 | 0 | 0 | 30.0 | 100.0 | 16.7 | 0.0 | 0.0 |
| Cell 6 | 1 | 660 | 0 | 0 | 0 | 0 | 65.0 | 0.0 | 0.0 | 0.0 | 0.0 |
| | Baseline | 420 | 10 | 120 | 120 | 53 | 0.0 | 100.0 | 100.0 | 100.0 | 100.0 |
| | 0.01 | 460 | 10 | 120 | 120 | 37 | 9.5 | 100.0 | 100.0 | 100.0 | 69.8 |
| | 0.1 | 540 | 10 | 120 | 77 | 24 | 28.6 | 100.0 | 100.0 | 64.2 | 45.3 |
| Cell 7 | 1 | 680 | 0 | 0 | 17 | 4 | 61.9 | 0.0 | 0.0 | 14.2 | 7.5 |
| | Baseline | 1950 | 10 | 120 | 120 | 120 | 0.0 | 100.0 | 100.0 | 100.0 | 100.0 |
| | 0.01 | 1900 | 10 | 120 | 120 | 120 | –2.6 | 100.0 | 100.0 | 100.0 | 100.0 |
| | 0.1 | 2000 | 10 | 120 | 120 | 120 | 2.6 | 100.0 | 100.0 | 100.0 | 100.0 |
| Cell 8 | 1 | 3000 | 0 | 0 | 0 | 0 | 53.8 | 0.0 | 0.0 | 0.0 | 0.0 |
| Cell 9 | Baseline | 1250 | 10 | 120 | 120 | 57 | 0.0 | 100.0 | 100.0 | 100.0 | 100.0 |
| | 0.01 | 1250 | 10 | 120 | 120 | 23 | 0.0 | 100.0 | 100.0 | 100.0 | 40.4 |
| | 0.1 | 1300 | 10 | 120 | 93 | 18 | 4.0 | 100.0 | 100.0 | 77.5 | 31.6 |
| | 1 | 1250 | 10 | 120 | 81 | 7 | 0.0 | 100.0 | 100.0 | 67.5 | 12.3 |

*Table 4 continued on next page*

Table 4 continued

| | [Drug] µM | Rheobase (pA) | Number of APs | | | | % change | Remaining AP (%) | | | |
|---|---|---|---|---|---|---|---|---|---|---|---|
| | | | 0.1 Hz | 1 Hz | 3 Hz | 10 Hz | Rheobase | 0.1 Hz | 1 Hz | 3 Hz | 10 Hz |
| | Baseline | 3200 | 10 | 120 | 120 | 60 | 0.0 | 100.0 | 100.0 | 100.0 | 100.0 |
| | 0.01 | 3250 | 10 | 120 | 111 | 51 | 1.6 | 100.0 | 100.0 | 92.5 | 85.0 |
| | 0.1 | 3250 | 10 | 120 | 107 | 40 | 1.6 | 100.0 | 100.0 | 89.2 | 66.7 |
| Cell 10 | 1 | 4700 | 10 | 59 | 43 | 34 | 46.9 | 100.0 | 49.2 | 35.8 | 56.7 |
| | | | | | | Baseline | 0.0 | 100.0 | 100.0 | 100.0 | 100.0 |
| | | | | | | 0.01 | 4.9 | 100.0 | 99.9 | 82.9 | 58.3 |
| | | | | | | 0.1 | 13.5 | 100.0 | 84.8 | 63.7 | 46.3 |
| | | | | | Average | 1 | 49.1 | 50.0 | 34.9 | 24.3 | 17.8 |
| | | | | | | Baseline | 0.0 | 0.0 | 0.0 | 0.0 | 0.0 |
| | | | | | | 0.01 | 3.1 | 0.0 | 0.1 | 9.3 | 10.7 |
| | | | | | | 0.1 | 5.1 | 0.0 | 9.4 | 12.5 | 11.0 |
| | | | | | SEM | 1 | 10.3 | 16.7 | 15.0 | 10.9 | 10.2 |

the membrane environment (Y1H), facing the protein-protein interface environment (M6norLeu), and facing the water-soluble environment (S11K and E12D). Rosetta also confirmed previously reported ProTx-II substitutions facing the membrane environment (Y1Q and W7Q, reported by Janssen *Flinspach et al., 2017*), facing the protein core environment (M19L, reported by Janssen *Flinspach et al., 2015*; *Neff and Wickenden, 2021*), and facing the protein-protein interface environment (K28E, reported by Amgen *Moyer et al., 2018*). We designed a novel and extensive hydrogen-bonding network at the ProTx-II – hNa$_V$1.7 VSD-II interface involving R20, R22, and E28 (on ProTx-II) and D816 (on hNa$_V$1.7 VSD-II) that contributed to improvements in the lead PTx2-3127 and PTx2-3258 peptide selectivity because Aspartate at position D816 is only present in hNa$_V$1.7 and hNa$_V$1.6 (see *Figure 2—figure*

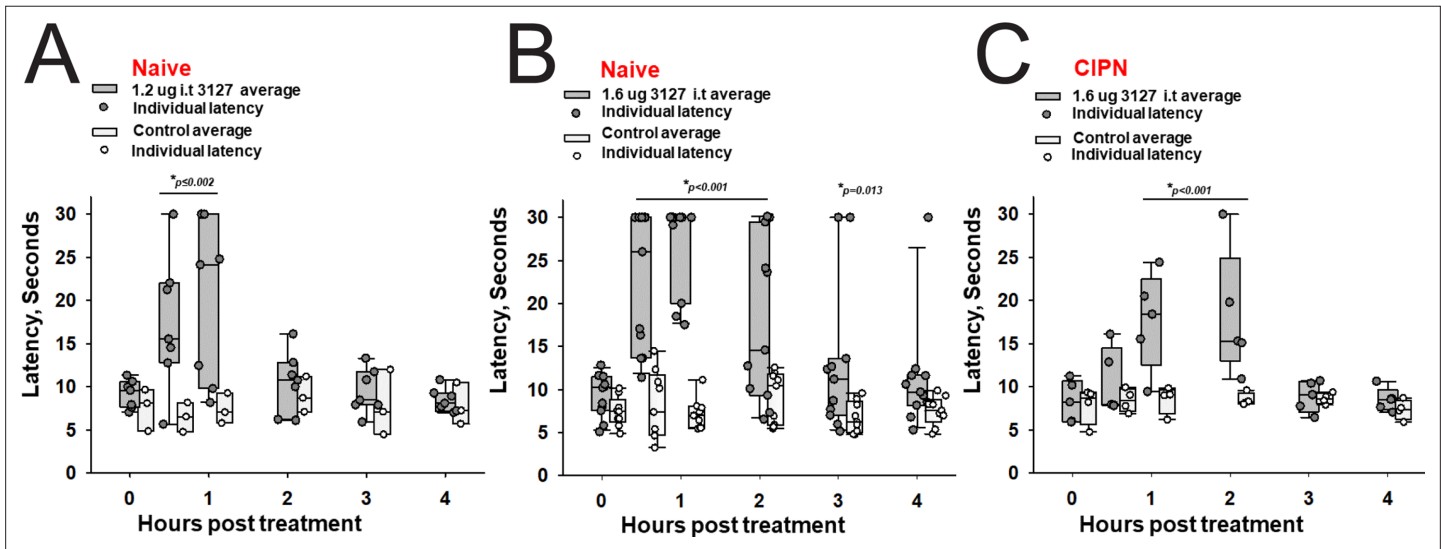

**Figure 8.** Efficacy of PTx2-3127 on thermal pain and CIPN neuropathy. PTx2-3127 exhibited dose dependent analgesia on a 52.1°C hotplate increasing the duration of effect as well as number reaching the latency cutoff with doses of 1.2 ug i.t. (**A**) to 1.6 ug i.t. (**B**) in naïve female and male rats. The analgesia mediated by PTx2-3127 was significant compared with vehicle controls for both doses (1.2 ug, p≤0.002) and the 1.6 ug dose several rats reached the hotplate latency cutoff (30 s to prevent injury) for several hours' duration (1.6 ug, p<0.001 and p=0.013 at indicated time points). (**C**) PTx2-3127 was also effective against oxaliplatin chemotherapy induced neuropathic pain (CIPN) with responses also significant compared with vehicle controls (p<0.001) and reaching the latency cutoff. (A–C, Two Way Repeated Measures ANOVA, Holm-Sidak method post hoc, treated versus control).

**Table 5.** Summary of four rounds of ProTx-II peptide optimization.

ProTx-II mutations that resulted in the most potent and selective peptide are highlighted in green. ProTx-II mutations that did not result in the most potent and selective peptide are highlighted in yellow. X at residue #22 in PTx2-2955 represents norArg.

| Residue # | 1 | 2 | 3 | 4 | 5 | 6 | 7 | 8 | 9 | 10 | 11 | 12 | 13 | 14 | 15 | 16 | 17 | 18 | 19 | 20 | 21 | 22 | 23 | 24 | 25 | 26 | 27 | 28 | 29 | 30 |
|---|---|---|---|---|---|---|---|---|---|---|---|---|---|---|---|---|---|---|---|---|---|---|---|---|---|---|---|---|---|---|
| WT PTx2 | Y | C | Q | K | W | M | W | T | C | D | S | E | R | K | C | C | E | G | M | V | C | R | L | W | C | K | K | K | L | W |
| PTx2-2955 | Y | C | Q | K | A | F | W | T | C | D | S | E | R | K | C | C | E | G | L | R | C | X | L | W | C | R | K | E | L | W |
| PTx2-3066 | Y | C | Q | K | W | M | Q | T | C | D | K | D | R | K | C | C | E | G | L | R | C | R | L | W | C | R | K | E | L | L |
| PTx2-3127 | Q | C | Q | K | W | M | Q | T | C | D | K | D | R | K | C | C | E | G | F | R | C | R | L | W | C | R | K | E | L | L |
| PTx2-3258 | H | C | Q | K | W | M | Q | T | C | D | K | D | R | K | C | C | E | G | F | R | C | R | L | W | C | R | K | E | L | L |

*supplement 1*). Our top designed peptides are highly potent, PTx2-3127 (IC$_{50}$=6.9 nM) and PTx2-3258 (IC$_{50}$=3.8 nM), and highly selective for hNa$_V$1.7 versus other human Na$_V$ channels. PTx2-3127 has 730-, 1671-, and 19,868-fold selectivity for hNa$_V$1.7 versus hNa$_V$1.2, hNa$_V$1.4 and hNa$_V$1.5, respectively. PTx2-3258 has 894-, 2336-, and 10,082-fold selectivity for hNa$_V$1.7 versus hNa$_V$1.2, hNa$_V$1.4, and hNa$_V$1.5, respectively. However, the potency and selectivity of our top peptides, PTx2-3127 and PTx2-3258, are superior to Janssen's JNJ63955918 peptide (*Flinspach et al., 2017*), underscoring the power of our structure-guided optimization approach.

Endogenous Na$_V$ channels do not always share the same pharmacology as recombinantly expressed channels, which could result from differences due to endogenous auxiliary subunits, interacting partners, or posttranslational modifications in native cells absent in heterologous systems (*Zhang et al., 2013*). Here, we demonstrate the efficacy of PTx2-3127 in targeting endogenous Na$_V$1.7 channels from both mice (*Figure 6*) and human neurons (*Figure 7*). Experiments in genetically identified mouse nonpeptidergic nociceptors show that PTx2-3127 inhibits the endogenous fast inward conductance consistent with Na$_V$1.7. PTx2-3127 also inhibits excitability and action potential firing in these neurons. These results suggest that mice have value as a preclinical model for developing PTx2 derivatives as pain therapeutics. Furthermore, in human DRG neurons treated with the chemotherapeutic oxaliplatin, PTx2-3127 also reduced neuronal excitability. Oxaliplatin is a chemotherapeutic agent commonly used to treat colorectal cancers (*Graham et al., 2004*) and is known to increase sensory neuron excitability to induce both neuropathic mechanical and cold allodynia. However, there is a debate regarding the mechanisms and Na$_V$ subtypes through which this occurs. We provide evidence that (1) PTx2-3127 can reduce action potential discharges in an in vitro model of oxaliplatin-induced neuropathy, and (2) Na$_V$1.7 contributes to human neuronal hyperexcitability in this model. This important result is consistent with PTx2-3127 retaining activity against endogenous Na$_V$ channels. We note that 1 µM PTx2-3127 partially inhibits hNa$_V$1.6 in HEK cells, and other hNa$_V$s to a lesser extent (*Table 1*), raising the possibility of off-target Navs contributing to neuronal modulation by 1 µM PTx2-3127. Overall, these results suggest that engineered PTx2 variants have potential to suppress human nociception in the clinic. The dual efficacy of PTx2-3127 in murine and human DRG neurons also demonstrates the value of combining the genetic power of mouse models with the translational relevance of in vitro experiments on human DRG neurons to validate future PTx2 variants during the preclinical optimization (*Shiers et al., 2020*).

PTx2-3127 demonstrated acute analgesia in keeping with a previous report of Na$_V$ channel blockade with ProTx-II and other targeted Na$_V$1.7 blocking approaches (*Flinspach et al., 2017*). Intrathecal administration of the peptide to otherwise naïve rats blocked pain sensitivity in the suprathreshold hotplate assay over a duration of several hours. After this hotplate assay rats were placed into an open field apparatus where the animals were ambulatory and explorative despite reaching a latency cutoff on the hotplate. The open field was not quantified in this setting because of the supra-stimulation of the hotplate assay directly preceding it. However, our observations correlate with published reports that Na$_V$ channel blockade in preclinical models parallels the human genetic mutant phenotype of pain insensitivity without motor function decrements (*Gingras et al., 2014*). Importantly PTx2-3127 was also effective against CIPN-induced neuropathy with intrathecal administration (*Figure 8*). The chronic pain of CIPN is often difficult to treat, but using the 52.1°C hotplate we demonstrated potent analgesia using PTx2-3127 in this model. It will be essential to characterize the expression levels and contribution of Na$_V$1.7 versus Na$_V$1.8 to the CIPN pain, as well as the pain modality (i.e. heat/cold, mechanical or chemical stimuli) where this highly selective Na$_V$1.7 blocking peptide is the most potent, however, in this initial investigation, intrathecal PTx2-3127 administration resulted in significant analgesia.

In summary, our interdisciplinary approach demonstrates the power of structure-guided peptide design and represents a major step toward the efficient development of potent and selective natural peptide-based inhibitors of human Na$_V$1.7 channels as prototypes of analgesic drug candidates for treating chronic pain. Additional work will be necessary to address the following limitations of our study and translate Na$_V$1.7 targeted peptides to the clinic. First, a significant improvement in the selectivity of our lead peptides for hNa$_V$1.7 versus hNa$_V$1.6 is needed to avoid affecting the function of motor neurons within a therapeutic concentration range (*Schmalhofer et al., 2008*). Second, optimization of the duration of peptide efficacy in vivo beyond several hours will be necessary to prolong the therapeutic effect in the clinic, either through increasing stability or continuous

administration via an intrathecal pump. Third, the development of peptide formulations will be useful to potentially enable peptide bioavailability through subcutaneous, intranasal or oral administration routes as are already in clinical use for GLP-1 (Glucagon-Like Peptide 1) receptor agonists (*Drucker, 2020*).

# Methods

**Key resources table**

| Reagent type (species) or resource | Designation | Source or reference | Identifiers | Additional information |
|---|---|---|---|---|
| Cell line (*Homo sapiens*) | HEK 293 | ATCC | Cat #: CRL-1573 | |
| Software, algorithm | Rosetta | Rosetta https://doi.org/10.1038/s41592-020-0848-2 https://www.rosettacommons.org/ | Version 3.12 | |
| Software, algorithm | IgorPro | IgorPro https://www.wavemetrics.com/ | Version 8 | |
| Software, algorithm | UCSF Chimera | UCSF Chimera https://www.cgl.ucsf.edu/chimera/ | Version 1.16 | |
| Software, algorithm | CHARMM-GUI | CHARMM-GUI https://doi.org/10.1002/jcc.20945 http://www.charmm-gui.org | Version 3.0 | |
| Software, algorithm | CHARMM36 | CHARMM36 https://doi.org/10.1002/jcc.23354 http://mackerell.umaryland.edu/charmm_ff.shtml | Version July 2019 | |
| Software, algorithm | Pulse-PulseFit | Pulse-PulseFit (HEKA Electronik GmbH, Germany) http://www.heka.com/index.html | Version 8.8 | |
| Software, algorithm | Origin | https://www.originlab.com/ | Version 9.0 | |

## Design of ProTx-II based peptides targeting hNav1.7

### Molecular dynamics simulation of ProTx-II - NavAb/Nav1.7 chimera complex

We ran a molecular dynamics simulation of the cryo-EM structure of NavAb/Nav1.7 in a complex with ProTx-II in a deactivated state (PDB ID: 6N4R) (*Xu et al., 2019*) to obtain a closer look at the interaction of ProTx-II with lipid membrane at the residue level. CHARMM-GUI (*Jo et al., 2008*) was used to embed the structure in a lipid bilayer of POPC with explicit TIP3P water molecules at a concentration of 150 mM NaCl. The system contained approximately 90,000 atoms and was parameterized with the CHARMM36 forcefield (*Huang and MacKerell, 2013*). Neutral pH was used to assign the protonation state as default, and the C-terminal of ProTx-II is in the amidated form. The simulation was run on our local GPU cluster using NAMD version 2.12 (*Jiang et al., 2011*). After 10,000 steps of steepest descent minimization, 1 fs timestep equilibrations were started with harmonic restraints initially applied to protein-heavy atoms and lipid tail dihedral angles as suggested by CHARMM-GUI (*Jo et al., 2008*). These restraints were gradually released over 2 ns. Harmonic restraints (0.1 kcal/mol/Å$^2$) were applied only to protein backbone heavy atoms. The systems were equilibrated further for 20 ns using 2 fs timestep with all bonds to hydrogen atoms constrained by the SHAKE algorithm (*Ryckaert et al., 1977*). The equilibrations were performed in NPT ensemble with semi-isotropic pressure coupling to maintain the correct area per lipid, and a constant temperature of 303.15 K. Particle Mesh Ewald (PME) method was used to compute electrostatic interactions. Non-bonded pair lists were updated every 10 steps with a list cutoff distance of 16 Å and a real space cutoff of 12 Å with energy switching starting at 10 Å. The production run was conducted for 100 ns without applied protein backbone restraints.

We analyzed the 100 ns simulation for interactions of ProTx-II residues with the surrounding environment, categorized into different groups: lipid head, lipid tail, water and VSDII (hNav1.7/NavAb chimera structure (PDB ID: 6N4R) *Xu et al., 2019*). Fractional contact is defined as the frequency of forming contact (3.5 Å as a cutoff) of heavy atoms belonging to the associated groups normalized over the course of simulation and across interacting chains, A-E, B-F, C-G, D-H of the structure.

## Computational design of ProTx-II variants

First, the cryo-EM structure of ProTx-II in complex with hNav1.7/NavAb in a deactivated state (PDB ID: 6N4R) was further refined in Rosetta (*Leman et al., 2020*) using Rosetta cryo-EM refinement protocol (*Dustrude et al., 2016*) (see the Methods section below entitled 'Rosetta Scripts for refinement of ProTx-II - hNav1.7/NavAb complex'). We generated 1000 refined models and extracted the top 10 scoring models for visual inspection. We carefully examined how well ProTx-II fits into the electron density across multiple interacting chains A-E, B-F, C-G, and D-H of the top models and eventually selected chain A-E for the subsequent modeling.

Rosetta FastDesign (*Maguire et al., 2021*) was used to introduce ProTx-II substitutions and design new peptide variants. A small deviation of backbone conformation is inherently sampled in FastDesign by ramping cycles of reduced repulsive forces. We seek to sample higher degrees of backbone flexibility during the design process by further incorporating Rosetta Small mover and Roll mover. Small mover performs small random changes in the backbone torsional space while Roll mover invokes small rigid body perturbation between ProTx-II and VSD-II. Both movers were implemented in Rosetta XML scripts prior to the FastDesign mover (see XML scripts in the Methods section below entitled 'Rosetta Scripts for refinement of ProTx-II - hNav1.7/NavAb complex').

FastDesign was used in conjunction with sequence profile constraints to control amino acid identity substitutions. In the computational design step (round 2), fixed identity was applied to positions that reflect empirical knowledge such as R20, R22, E28 for preserving the hydrogen bond network with D816 (VSD-II) and W5, M6, W24, R26, K27, L29 for forming important interactions with the channel as observed from the modeling results of prior designs. On top of that, we disallowed acidic residues for positions that have significant interactions with lipid heads or lipid tails observed from the fractional contacts derived from the MD simulation of the ProTx-II – hNav1.7/NavAb chimera. Other positions, except disulfides, were allowed to freely mutate. However, we used Rosetta FavorSequenceProfile mover to slightly bias new substitutions toward native residues on ProTx-II because of the lack of secondary structure elements for the majority of ProTx-II fold in combination with a higher degree of backbone flexibility could result in highly diverse set of amino acid substitutions with FastDesign. We generated 1000 designs and extracted the 100 top designs by total score followed by selecting the top 20 designs by Rosetta DDG. The consensus designed sequence was constructed from the top 20 designs using sequence logo presentation (see *Figure 3—figure supplement 1*) and analyzed in combination with available experimental data during each optimization round as described in the main text.

## Peptide synthesis and folding

The ProTx-II variants were produced synthetically using Fmoc automated solid-phase synthesis performed on Liberty Blue peptide synthesizer from CEM Inc using a microwave-assisted synthesis strategy employing diisopropyl carbodiimide and Oxyma for the activation chemistry. Pre-loaded ChemMatrix (Sigma Aldrich) Wang resins were used to produce ProTx-II variants with C-terminal acids. Acidolytic cleavage and deprotection of the completed peptide resins was performed with 9.5 ml trifluoroacetic acid (TFA), 0.5 ml $H_2O$, 0.5 ml Anisole, 0.5 ml thioanisole, 0.25 ml of DODT (3,6-dioxa-1,8-octanedithiol), 0.25 ml triisopropyl silane per gram of resin for 2 h at room temperature. Cleaved peptides were precipitated with 5-fold excess of diethyl ether added directly to the pre-filtered cleavage solution, isolated, and re-solubilized in TFA. Linear peptides were purified by preparative HPLC using a Phenomenex Luna C18(2), 100 Å pore size, 10 μ particle size, 250 mm x 21.2 mm column and a 15–48% linear gradient of acetonitrile with 0.05% TFA over 40 min. Molecular weights were confirmed by LC/MS and fractions were pooled for folding. Purified linear fractions were added directly to 20 mM Tris, 2 M Urea, 1:2 oxidized/reduced glutathione, and pH was adjusted to 7.8–8.0 using acetic acid. Final peptide concentration was approximately 0.1–0.2 mg/ml. Solutions were stirred for 24–48 h at room temperature. Folded peptides were purified using a Phenomenex Luna C18(2), 100 Å pore size, 10 μ particle size, 250 mm x 21.2 mm column with a 15–48% linear gradient of acetonitrile with 0.05% TFA over 40 min. Main peak fractions were analyzed by HPLC and LC/MS. Peptide fractions with a purity >95% were pooled, flash-frozen and subsequently lyophilized. Peptide content for each product was determined by absorbance at 280 nm using the calculated extinction coefficient. Percent purity was determined by HPLC using a Phenomenex Luna C18(2) analytical column, 250 mm x 4.6 mm, 100 Å pore size, 5 μ particle size. Peptide mass and oxidation

were confirmed by LC/MS using a Waters 2965 separations module coupled to a Waters Micromass ZQ electrospray mass spectrometer.

## Testing of designed peptides potency and selectivity using electrophysiological assays on recombinant channel cell lines

HEK-293 cells stably expressing human $Na_V1.1$, $Na_V1.4$, $Na_V1.5$, $Na_V1.6$, and $Na_V1.7$ were obtained from Dr. Chris Lossin. Rat $Na_V1.3$ expressing HEK-293 cells were from Dr. Stephen Waxman (Yale University, New Haven, CT). These cell lines were cultured in complete DMEM supplemented with 10% FBS, 1% penicillin/streptomycin, and G418. The human $Na_V1.8$ channel (co-expressing with human $Na_V\beta1$ and $Na_V\beta2$ subunits) and the $Nav1.9$ channel (co-expressing with human Trkb, $Na_V\beta1$, and $Na_V\beta2$ subunits) were obtained from Dr. Neil Castle (Icagen, Durham, NC). $hNa_V1.8$ cells cultured with G418 (0.4 mg/mL) and puromycin (0.5 ng/mL) and $hNa_V1.9$ cells were cultured with G418 (0.4 mg/mL), puromycin (0.5 ng/mL), and zeocin (0.05 mg/mL). Human $Na_V1.2$ were expressed transiently by transfection of the $hNa_V1.2$ cDNA (from Dr. Alan L. Goldin, UC Irvine, CA) into HEK-293 cells.

Whole-cell patch-clamp experiments on recombinant channels were conducted manually at room temperature (22–24°C) using an EPC-10 amplifier (HEKA Electronik, Lambrecht/Pfalz, Germany). Cells were trypsinized and plated onto poly-l-lysine–coated coverslips. All recordings were done in normal Ringer external bath solution containing (in mM) 160 NaCl, 4.5 KCl, 2 CaCl$_2$, 1 MgCl$_2$, 10 HEPES (pH 7.4 and 305 mOsm) as. Patch pipettes were pulled from soda lime glass (micro-hematocrit tubes, Kimble Chase, Rochester, NY) and had resistances of 2–3 MΩ when filled with CsF-based internal solution containing (in mM) 10 NaF, 110 CsF, 20 CsCl, 10 HEPES, 2 EGTA, (pH 7.4, 310 mOsm). Data acquisition and analysis were performed with Pulse-PulseFit (HEKA Electronik GmbH, Germany), IgorPro (WaveMetrics, Portland, OR), and Origin 9.0 software (OriginLab Corporation, Northampton, MA). Cells were held at −90 mV and voltage stepped to −120 mV for 200ms before depolarizing to −10 mV for 50 ms to elicit inward currents. Control test currents were monitored for 5–10 min to ensure that the amplitude and kinetics of the response were stable. Series resistance was compensated to 80–90% and linear leak currents and capacitance artefacts were corrected using a P/4 subtraction method. Pulse interval was 0.1 Hz and peptides were applied to individual cells using a glass transfer pipette directly into the recording bath. For measuring inhibition, currents were allowed to saturate with repeated pulsing before addition of subsequent doses. IC$_{50}$ values were derived from measurements performed on individual cells that were tested with at least three or more concentrations of each peptide. Concentration response curves were fitted with the Hill equation and IC$_{50}$s are reported with 95% confidence intervals.

## Testing of designed peptides stability in artificial cerebrospinal fluid

Stability In Artificial Cerebrospinal Fluid (aCSF): The stability of the wild-type ProTx-II, PTx2-3127, and PTx2-3258 was conducted in artificial Cerebrospinal Fluid (aCSF). The aCSF was purchased from Tocris Biosciences (Catalog # 3525) and had the following ionic composition (in mM): Na$^+$ 150; K$^+$ 3.0; Ca$^{2+}$ 1.4; Mg$^{2+}$ 0.8; P 1.0; Cl$^-$ 155. The wild-type ProTx-II, PTx2-3127, and PTx2-3258 were dissolved in DPBS at 200 µM (1 mg of respective peptides in 1.305 mL, 1.315 mL, and 1.315 mL of DPBS, respectively). 500 µL of dissolved peptide in DPBS and 1.500 mL of aCSF were mixed to get 50 µM peptide solution in aCSF. The samples were incubated at 37°C and aliquots of 100 µL were removed at 0, 1, 2, 4, 8, 12, 24, and 120 hr, respectively. The aliquots were immediately flash frozen and stored at –80°C until further analysis. Peptides dissolved in aCSF were analyzed on a Hewlett Packard 1100 series HPLC system and monitored at 214 nm and 280 nm. The stability at various time points was determined by calculating the average Area under the curve at 214 nm and 280 nm for 2 injections of 20 µL using ChemStation Software. The peptides were run on a BioBasic C18 column (150X4.8 mm, ThermoFisher). The mobile phases were 0.1% Trifluoroacetic Acid in water (mobile phase A) and 100% Acetonitrile (mobile phase B).

## Testing of designed peptides efficacy on mouse sensory neurons
### Mice
This study was approved by the UC Davis Institutional Animal Care and Use Committee and conforms to guidelines established by the NIH. Mice were maintained on a 12 hr light/dark cycle, and food

and water were provided ad libitum. The *Mrgprd^GFP^ mouse line was a generous gift from David Ginty (Harvard University, Boston MA) (MGI: 3521853).*

## Preparation of DRG sections

This study was approved by the UC Davis Institutional Animal Care and Use Committee and conforms to guidelines established by the NIH. 20-week-old *Mrgprd^GFP^* mice was briefly anesthetized with 3–5% isoflurane and then decapitated. The spinal column was dissected, and excess muscle tissue removed. The spinal column was then bisected in the middle of the L1 vertebrae identified by the 13th rib and drop fixed for 1 hr in ice cold 4% paraformaldehyde in 0.1 M phosphate buffer (PB) pH adjusted to 7.4. The spine was washed 3× for 10 min each in PB and cryoprotected at 4°C in 30% sucrose diluted in PB for 24 hr. The spine was cut into sections containing two vertebra per sample which were frozen in Optimal Cutting Temperature (OCT) compound (Fisher Cat#4585) and stored at –80°C until sectioning. Vertebrae position relative to the 13th rib was recorded for each frozen sample to determine the specific vertebrae position in the spinal cord. Samples were cut into 30 μm sections on a freezing stage sliding microtome and were collected on Colorfrost Plus microscope slides (Fisher Scientific Cat#12-550-19). Slides were stored at –20°C or immediately used for multiplex immunofluorescence labeling.

## Multiplex immunofluorescence labeling

A hydrophobic barrier was drawn around tissue sections mounted on slides as described above using a hydrophobic barrier pen (Scientific Device Cat#9804–02). Sections were incubated in 4% milk in PB containing 0.2% Triton X-100 (vehicle) for 1 hr and then incubated in vehicle containing 0.1 mg/mL IgG F(ab) polyclonal IgG antibody (Abcam cat# ab6668) for 1 hr. Sections were washed 3× for 5 min each in vehicle and then incubated in vehicle containing primary Abs. (Supplemental Table Abs) for 1 hr. Sections were washed 3× for 5 min each in vehicle and then incubated in vehicle containing mouse IgG-subclass-specific goat secondary Abs (Table Abs) conjugated to Alexa Fluors (Thermo Fisher). Sections were washed 3× for 5 min each in PB and mounted with Prolong Gold (Thermo Fisher) and Deckglaser cover glass (Cat#NC1776158). All incubations and washes were done at room temperature with gentle rocking.

## Immunofluorescence imaging

Images were acquired with an inverted scanning imaging system (Zeiss LSM 880, 410900-247-075) run by ZEN black v2.1. Laser lines were 488 nm, 633 nm. Low-magnification images were acquired in confocal mode with a 0.8 NA 20 x objective (Zeiss 420650–9901) and reconstructed as a tiled mosaic using ImageJ. High-magnification images were acquired in airy disk imaging mode with a 1.4 NA 63 x oil objective (Zeiss 420782-9900-799). Linear adjustments to contrast and brightness and average fluorescence intensity z-projections were performed using ImageJ software.

## Neuron cell culture

Cervical, thoracic and lumbar DRGs were harvested from 4- to 6-week-old *MrprD-GFP* mice and transferred to Hank's buffered saline solution (HBSS) (Invitrogen). Ganglia were treated with collagenase (2 mg/ml; Type P, Sigma-Aldrich) in HBSS for 15 min at 37°C followed by 0.05% Trypsin-EDTA (Gibco) for 2.5 min with gentle rotation. Trypsin was neutralized with culture media (MEM, with L-glutamine, Phenol Red, without sodium pyruvate) supplemented with 10% horse serum (heat-inactivated; Gibco), 10 U/ml penicillin, 10 μg/ml streptomycin, MEM vitamin solution (Gibco), and B-27 supplement (Gibco). Serum-containing media was decanted and cells were triturated using a fire-polished Pasteur pipette in MEM culture media containing the supplements listed above. Cells were plated on laminin-treated (0.05 mg/ml, Sigma-Aldrich) 5 mm Deckglaser coverslips, which had previously been washed in 70% ethanol and UV-sterilized. Cells were then incubated at 37°C in 5% $CO_2$. Cells were used for electrophysiological experiments 24–38 hr after plating.

## Voltage clamp of endogenous neuronal sodium channels

Voltage clamp was achieved with a dPatch amplifier (Sutter Instruments) run by Sutterpatch (Sutter Instruments). Solutions for voltage-clamp recordings: internal (in mM) 15 NaCl, 100 CsCl, 25 CsF,

1 EGTA and 10 HEPES adjusted to pH 7.3 with CsOH, 297 mOsm. Seals and whole-cell configuration were obtained in an external patching solution containing the following (in mM) 145 NaCl, 3.5 KCl, 1.5 $CaCl_2$, 1 $MgCl_2$, 10 HEPES, 10 Glucose adjusted to pH 7.4 with NaOH, 322 mOsm. For voltage-clamp neuronal recordings, the external solution contained (in mM) 44 NaCl, 106 TEA-Cl, 1.5 $CaCl_2$, 1 $MgCl_2$, 0.03 $CdCl_2$ 10 HEPES, 10 glucose, pH adjusted to 7.4 with TEA-OH, 315 mOsm. The calculated liquid junction potential for the internal and external recording solutions was 5.82 mV and not accounted for. Osmolality is measured with a vapor pressure osmometer (Wescor, 5520). For voltage-clamp recordings, neurons plated on the cover glass as described in the *Neuron Cell Culture* section were placed in a recording chamber (Warner Cat#64–0381) and were rinsed with an external patching solution using a gravity-driven perfusion system. Neurons from *Mrgprd*[GFP] mice showing intracellular GFP were then selected for patching. After the whole-cell voltage clamp was established the external patching solution was exchanged with the external recording solution using a gravity-driven perfusion system. PTx2-3127, vehicle control (external recording solution) and TTX were kept on ice and diluted in room temperature (20–22°C) external recording solution just prior to application to neurons and manually added at a rate of approximately 1 mL/min. Experimenter was blinded to the identity of PTx2-3127 versus vehicle control solutions during recordings. PTx2-3127, vehicle control and TTX were applied to neurons using separate perfusion lines to prevent contamination. After each neuron, perfusion lines were cleared with 1 mL of 70% ethanol followed by 1 mL of milli Q water and were then filled with an external recording solution. Thin-wall borosilicate glass recording pipettes (BF150-110-10, Sutter) were pulled with blunt tips, coated with silicone elastomer (Sylgard 184, Dow Corning), heat cured, and tip fire-polished to resistances less than 3 $M\Omega$. Series resistance of 3–8 $M\Omega$ was estimated from the whole-cell parameters circuit. Series resistance compensation between 37 and 77% was used to constrain voltage error to less than 15 mV, lag was 6 μs. Cell capacitances were 13–34 pF. Capacitance and Ohmic leak were subtracted using a P/4 protocol. Output was low-pass filtered at 10 kHz using the amplifier's built-in Bessel and digitized at 50 kHz. The average current in the initial 0.14 s at holding potential prior to the voltage step was used to zero-subtract each recording. The mean current was the current amplitude between 0.4–1ms into the 0 mV step. Peak current amplitude was the peak current amplitude between 0.4 and 8 ms into the 0 mV step. Experiments were performed on neurons with membrane resistance greater than 1 $G\Omega$ assessed prior to running voltage clamp or current clamp protocols while neurons were held at a membrane potential of –80 mV. Data with predicted voltage error, $V_{error} \geq 15$ mV were excluded from the analysis. $V_{error}$ was tabulated using estimated series resistance post compensation and peak $Na_V$ current.

## Current clamp

Solutions for current clamp recordings: internal (in mM) 120 K-methylsulfonate, 10 KCl, 10 NaCl, 5 EGTA, 0.5 $CaCl_2$, 10 HEPES, 2.5 MgATP, and adjusted to pH 7.2, 289 mOsm. External solution (in mM) 145 NaCl, 5 KCl, 2 $CaCl_2$, 2 $MgCl_2$, 10 HEPES, 10 Glucose adjusted to pH 7.3 with NaOH, 308 mOsm. The calculated liquid junction potential for these solutions was 9.7 mV which was not accounted for unless noted. Thin-wall borosilicate glass recording pipettes (BF150-110-10, Sutter) were pulled with blunt tips and tip fire-polished to resistances less than 3 $M\Omega$. For current-clamp recordings, neurons plated on the cover glass as described in the *Neuron Cell Culture* section were placed in a recording chamber (Warner Cat#64–0381) and were rinsed with external solution using a gravity-driven perfusion system. Neurons from *Mrgprd*[GFP] mice showing intracellular GFP were then selected for patching. The same protocol for application of PTx2-3127, vehicle control (external solution) and TTX described in the *Voltage Clamp* section was followed. In current clamp experiments data were excluded if the resting membrane potential of a neuron rose above –40 mV. After adjusting for the predicted liquid junction potential offset, the resting membrane potential of neurons in *Figure 6* ranged from –57 to –78 mV and the resting membrane potential of TTX-insensitive neurons in *Figure 6—figure supplement 1* ranged from –54 to –70 mV.

## Experimental design and statistical treatment

Independent replicates (*n*) are individual neurons from multiple mice, details in figure legends. Statistical tests were conducted using Igor 8 (Wavemetrics Inc), details in figure legends.

## Testing of designed peptides efficacy on human sensory neurons

All human tissues that were used for the study were obtained by legal consent from organ donors in the US. AnaBios Corporation's procurement network includes only US based Organ Procurement Organizations and Hospitals. Policies for donor screening and consent are the ones established by the United Network for Organ Sharing (UNOS). Organizations supplying human tissues to AnaBios follow the standards and procedures established by the US Centers for Disease Control (CDC) and are inspected biannually by the DHHS. Distribution of donor medical information is in compliance with HIPAA regulations to protect donor's privacy. All transfers of donor tissue to AnaBios are fully traceable and periodically reviewed by US Federal authorities. AnaBios generally obtains donor organs/tissues from adults aged 18–60 years old. Donor DRGs from males and females were harvested using AnaBios' proprietary surgical techniques and tools and were shipped to AnaBios via dedicated couriers. The DRGs were then further dissected in cold proprietary neuroplegic solution to remove all connective tissue and fat. The ganglia were enzymatically digested, and the isolated neurons put in culture in DMEM F-12 (Gemini Bio-Products CAT#: 900–955. Lot# M96R00J) supplemented with Glutamine 2 mM, Horse Serum 10% (Invitrogen #16050–130), hNGF (25 ng/ml) (Cell Signaling Technology #5221LF), GDNF (25 ng/ml) (ProSpec Protein Specialist #CYT-305) and Penicillin/Streptomycin (Thermo Fischer Scientific #15140–122).

External Current Clamp solution included: 145 mM NaCl, 3 mM KCl, 1 mM $MgCl_2$, 2 mM $CaCl_2$, 10 mM dextrose, 10 mM HEPES, pH = 7.4 (with NaOH), 300±5 mOsm. Internal Current Clamp solution included: 110 mM $K^+$ gluconate, 20 mM KCl, 10 mM EGTA, 8 mM NaCl, 4 mM Mg-ATP, 10 mM HEPES, pH = 7.3 (with KOH), 280±5 mOsm. All of our compounds come from Sigma-Aldrich. PTx2-3127 was stored in 10 mM formulation in DMSO at –20°C. Oxaliplatin was stored in 50 mM formulation in DMSO at 4°C.

DRG recordings were obtained from human DRG in culture (2–7 days). Human DRG neurons were incubated with Oxaliplatin (50 µM) at 37 °C for 24 hr. Whole-cell patch-clamp recordings were conducted under current-clamp mode at room temperature (~23°C) using HEKA EPC-10 amplifier. Data were acquired on a Windows-based computer using the PatchMaster program. Pipettes (1.5–3.0 MΩ) (Warner Instruments #64–0792) were fabricated from 1.5 mm borosilicate capillary glass using a Sutter P-97 puller. Cells on Corning glass coverslips (Thomas Scientific #354086) were transferred to a RC-26GLP recording chamber (Warner Instruments #64–0236) containing 0.5 ml standard external solution. Extracellular solution exchange was performed with rapid exchange perfusion system (flow rate 0.5–1 ml/min) (Warner Instruments #64–0186). Cells for recordings were selected based on smoothness of the membrane. Cells were held at a resting membrane potential. Signals were filtered at 3 kHz, sampled at 10 kHz. Once whole-cell access was obtained the cell was allowed an equilibration time of at least 5 min. Once the cell under recording stabilized, rheobase of single action potentials were assessed. Action potentials were induced by a train of 10 individual current steps 20ms in. duration, delivered at 0.1 Hz and 120 individual current steps delivered at 1, 3, and 10 Hz, using current injection at 150% of rheobase of baseline. Test compound concentrations were washed in for 5 min and step 6 and 7 were repeated for each concentration. Exclusion criteria: series resistance >15 MΩ; unstable recording configuration (15% change of rheobase or access resistance within the same concentration); time frame of drug exposure not respected.

The percentage of action potentials remaining was calculated as the number of action potentials divided by the number of action potentials obtained under control condition at the same frequency. One-way ANOVA (SigmaPlot v14) with Tukey, Bonferroni and Dunnett post-hoc test was used to determine the significance of difference between treatment and control (as specified in the figure and table legends).

## Testing of designed peptides efficacy in animal models of pain

### Animals

All experiments using live animals were conducted in accordance with protocols approved by the Institutional Animal Care and Use Committee of the University of California and adhered to the National Institutes of Health guide for the care and use of Laboratory animals. Great care was taken to reduce the number and minimize suffering of the animals used. Sprague–Dawley male and female rats (250–300 g; Charles River, Wilmington, MA, USA) were housed with free access to food and water. They were maintained under a 12 h light/dark cycle with controlled temperature and relative humidity.

After acclimation, the animals were each assayed for their baseline responses and then a day later received an intrathecal port placement. After recovery from the port surgery, the rats were assessed for post-surgery behavioral testing. For peptide treatments, rats were randomly divided into groups and tested with assays performed between 9:00 a.m. and 5:00 p.m. Scientists running the experiments were blinded to the treatment protocol at the time of the tests.

For the intrathecal cannulation briefly, the rats were anesthetized by isoflurane inhalation and the hair on the back at the surgical site shaved and the skin cleaned with ethyl alcohol and betadine per aseptic technique and incised about 1 cm in length. The muscle on the side of the L4 -L5 vertebrae was incised and retracted to place a catheter into the subarachnoid space. The tissue was incised by the tip of a bent needle, which allows escape of a small amount of cerebral spinal fluid (CSF). The caudal edge of the cut is lifted, and an intrathecal catheter, 32ga (0.8Fr) PU 18 cm, fixed to a stylet with a 27ga luer stub (Instech Laboratories) was gently inserted into the intrathecal space in the midline, dorsal to the spinal cord. The catheter was inserted coinciding with the placement of the distal end of the catheter in proximity to the spinal cord the lumbar vertebrae. The exit end of the catheter is taken out through an opening in the skin and connected to an access port. Rats received 2 mg/kg meloxicam once post surgically and 1 mg/kg daily up to 48 hr post-surgery if needed. The rats were allowed to recover for 7 days and then motor activity of the rats was examined for any sign of alteration. Competent rats were then randomly assigned to groups and tested with experimental compounds and assessed in behavioral assays. At necropsy after the end of the experiments, catheter placements were ensured by injection of colored dye any nonpatent catheters were excluded from the results.

Chemicals: the peptides were stored at –20°C in dry powder. The powder was weighed on an analytical balance and an amount of sterile artificial cerebral spinal fluid (ACSF, Fischer Scientific) was added to formulate concentrations of 1 mg/mL stock which was diluted to the desired concentration for each individual experiment. Stock solutions were aliquoted and stored at –20°C until further use. Peptide solutions were delivered with a Hamilton airtight syringe fit with an autoinjector (Instech laboratories) and 10 μL volume of the selected concentration or ACSF vehicle was injected intrathecally via the cannula and followed by 100 μL ACSF. The treatments were randomized to include different treatments and controls within the same day experimental setting and observers were blinded to the treatments.

Behavioral assays: on the test day animals were first tested for their baseline score in the open field and then hotplate. The open field assay was conducted in an open-field arena (40Wx40 L x 30H cm) of a 16-square grid clear acrylic open top chamber. Behavior and activity were monitored for 2 min. Activity was assessed by the number of lines each animal crosses with both hind paws and number of rears as a function of time. The purpose of the open field was to ensure there was not a significant change in motor skill due to the cannulation surgery. Open field ambulatory activity was assessed after long hotplate latency in some animals, but it was not quantified as a treatment outcome given the high stimulated state after the nociceptive tests and the difference in duration on the hotplate between treatment and control groups. Thermal nociceptive assay: The thermal nociception was assessed with a hotplate plate with the intensity set at a constant 52.1°C. Animals were placed individually on the warm metal surface and timed until their response of hind paw licking or jumping. A cutoff time limit of 30 s was imposed to prevent tissue damage. After paw licking or jump behavior is observed rats were immediately removed from the hotplate. One trial was used for baseline and timepoint assessment in order to not overstimulate or train the animals to the stimulus. Limiting exposure to the hotplate also ensured that no tissue damage occurred with animals that reached the cutoff.

Chronic pain models: Chemotherapy induced neuropathy was induced in rats with oxaliplatin after i.t catheter placement recovery with a single i.p. dose of oxaliplatin 6 mg/kg. The animals were allowed to recover for 3 days and then were assessed in the open field assay to ensure motor function and with a von Frey assay to assess allodynia to verify their pain state. The von Frey assay with an electronic aesthesiometer quantified the average baseline for a group of male and female rats to be 72.9±2.7 grams for the mechanical withdrawal threshold after cannulation but before CIPN model induction which fell to 27.9±2.7 grams indicating allodynia. On the day of treatment rats were assessed for baseline measures and then treated and assayed for thermal nociceptive responses.

## 1. Rosetta scripts for refinement of ProTx-II - hNav1.7/NavAb complex
### 1.1. Rosetta command lines

```
~Rosetta/main/source/bin/rosetta_scripts.linuxgccrelease \
    -database~Rosetta/main/database/ \
    -in::file::s $pdb \
    -parser::protocol $xml \
    -ignore_unrecognized_res \
    -edensity::mapreso 4.2 \
    -default_max_cycles 200 \
    -relax:constrain_relax_to_start_coords \
    -edensity::cryoem_scatterers \
    -use_input_sc \
    -beta \
    -missing_density_to_jump \
    -out::prefix EM-relax-density- \
    -crystal_refine \
    -nstruct 5
```

### 1.2. Rosetta XML scripts

```
<ROSETTASCRIPTS>
    <SCOREFXNS>
        <ScoreFunction name="beta" weights="beta_cart"/>
        <ScoreFunction name="dens" weights="beta_cart">
            <Reweight scoretype="elec_dens_fast" weight="35.0"/>
            <Set scale_sc_dens_byres="R:0.76,K:0.76,E:0.76,D:0.76,M:0.76,C:0.
81,Q:0.81,H:0.81,N:0.81,T:0.81,S:0.81,Y:0.88,W:0.88,A:0.88,F:0.88,P:0.88,I:
0.88,L:0.88,V:0.88"/>
        </ScoreFunction>
    </SCOREFXNS>
    <MOVERS>
        <SetupForDensityScoring name="setupdens"/>
        <LoadDensityMap name="loaddens" mapfile="../6N4R.mrc "/>
        <FastRelax name="relaxcart" ramp_down_constraints="false"
scorefxn="dens" repeats="2" cartesian="1"/>
    </MOVERS>
    <PROTOCOLS>
        <Add mover="setupdens"/>
        <Add mover="loaddens"/>
        <Add mover="relaxcart"/>
    </PROTOCOLS>
    <OUTPUT scorefxn="beta"/>
</ROSETTASCRIPTS>
```

## 2. Rosetta scripts for computational design of ProTx-II variants
### 2.1. Rosetta command lines

```
#!/bin/bash
if [ $# -lt 3 ]; then
     echo "USAGE: runDesign.sh <pdb> <xml> <resfile>"
     exit
fi
pdb=$1
xml=$2
```

```
resfile=$3
~Rosetta/main/source/bin/rosetta_scripts.macosclangrelease \
    -in:path:databas ~Rosetta/main/database \
    -in:file:fullatom \
    -in:file:s $pdb \
    -parser:protocol $xml \
    -parser:script_vars resfile=$resfile \
    -nstruct 20 \
    -linmem_ig 10 \         -optimization:default_max_cycles 200 \
    -out:file:scorefile score-design-$[resfile].sc \
    -out:prefix design-$[resfile]- \
    -overwrite
```

## 2.2. ProTx-II resfile

```
PIKAA ACDEFGHIKLMNPQRSTVWY
start
24 E NATAA
5 E NATAA
6 E NATAA
22 E NATAA
27 E NATAA
29 E NATAA
20 E PIKAA R
28 E PIKAA E
30 E PIKAA L
1 E NOTAA ED
4 E NOTAA ED
7 E NOTAA ED
8 E NOTAA ED
13 E NOTAA ED
```

## 2.3. Rosetta XML file

```
<ROSETTASCRIPTS>
    <SCOREFXNS>
        <ScoreFunction name="ref2015" weights="ref2015"/>
        <ScoreFunction name="ref2015_cst" weights="ref2015">
            <Reweight scoretype="coordinate_constraint" weight="1"/>
            <Reweight scoretype="atom_pair_constraint" weight="1"/>
            <Reweight scoretype="dihedral_constraint" weight="1"/>
            <Reweight scoretype="angle_constraint" weight="1"/>
            <Reweight scoretype="netcharge" weight="1.0" />
        </ScoreFunction>
        <ScoreFunction name="ref2015_cart" weights="ref2015_cart"/>
    </SCOREFXNS>
    <RESIDUE_SELECTORS>
        <Chain chains="E" name="peptide"/>        <Chain chains="A"
name="hNav"/>        <Neighborhood distance="8.0" name="peptide_
and_neighbors_8 A" selector="peptide"/>        <Neighborhood
distance="8.0" name="interface_hNav" selector="peptide"/>        <And
name="interface" selectors="peptide_and_neighbors_8 A,interface_
hNav"/>        <Not name="not_peptide_and_neighbors" selector="peptide_and_
neighbors_8 A"/>        <Index name="anchors" resnums="5,24"/>
    </RESIDUE_SELECTORS>
```

```
<TASKOPERATIONS>
    <InitializeFromCommandline name="init"/>        <ReadResfile
filename="%%resfile%%" name="rrf"/>        <RestrictChainToRepacking chain="2"
name="only_repack_chain"/>
    <DisallowIfNonnative disallow_aas="PCG" name="no_
PCG"/>        <OperateOnResidueSubset name="restrict_packing_to_hNav"
selector="hNav">
        <RestrictToRepackingRLT/>
    </OperateOnResidueSubset>
    <OperateOnResidueSubset name="prevent_to_not_peptide_and_neighbors"
selector="not_peptide_and_neighbors">
        <PreventRepackingRLT/>
    </OperateOnResidueSubset>
    <LimitAromaChi2 name="limchi2"/>
    <IncludeCurrent name="current"/>    </TASKOPERATIONS>
<FILTERS>
    <Ddg confidence="0" jump="1" name="ddg" repack="1"
repeats="5" scorefxn="ref2015" threshold="-20"/>
    <Ddg confidence="0" jump="1" name="ddg_norepack" repack="0"
repeats="1" scorefxn="ref2015" threshold="-20"/>
    <Sasa confidence="0" jump="1" name="interface_buried_sasa"/>
    <Sasa confidence="0" hydrophobic="True" jump="1" name="interface_
hydrophobic_sasa"/>
    <Sasa confidence="0" jump="1" name="interface_polar_sasa"
polar="True"/>
    <BuriedUnsatHbonds confidence="0" jump_number="1" name="BUH"
scorefxn="ref2015"/>
    <BuriedUnsatHbonds confidence="0" cutoff="1" ignore_surface_
res="true" name="new_buns_bb_heavy" print_out_info_to_pdb="true" report_
bb_heavy_atom_unsats="true" residue_selector="interface" residue_surface_
cutoff="20.0" scorefxn="ref2015"/>
    <BuriedUnsatHbonds confidence="0" cutoff="1" ignore_surface_
res="true" name="new_buns_sc_heavy" print_out_info_to_pdb="true" report_
sc_heavy_atom_unsats="true" residue_selector="interface" residue_surface_
cutoff="20.0" scorefxn="ref2015"/>
    <PackStat chain="1" name="Packstat" repeats="5" threshold="0.6"/>
    <InterfaceHbonds jump="1" name="interface_Hbonds" scorefxn="ref2015"
threshold="0"/>
</FILTERS>
<MOVERS>
    <AddConstraints name="add_hNav_constraints" >
      <CoordinateConstraintGenerator name="gen_csts" sd="0.1"
sidechain="false" native="false" residue_selector="hNav" />
    </AddConstraints>
      <ClearConstraintsMover name="clear_all_constraints"/>
      <FastDesign cartesian="0" name="design"
ramp_down_constraints="false" repeats="5" scorefxn="ref2015_cst"
task_operations="init,rrf,prevent_to_not_peptide_and_neighbors,only_repack_
chain,no_PCG,limchi2,current">
          <MoveMap bb="0" chi="0" jump="1" name="movemap_design">
              <ResidueSelector bb="1" chi="1" selector="peptide_and_
neighbors_8 A"/>
          </MoveMap>
      </FastDesign>
      <RollMover name="roll" start_res="1" stop_res="30"
random_roll="1" random_roll_angle_mag="0.15"
```

```
random_roll_trans_mag="0.35" />          <Small name="small" residue_
selector="peptide" scorefxn="ref2015_cst" nmoves="20"/>
            <FavorSequenceProfile name="favournative" weight="1.2" use_
current="true" matrix="IDENTITY"/>
    </MOVERS>
    <PROTOCOLS>
        <Add mover="add_hNav_constraints"/>
        <Add mover="roll"/>
        <Add mover="small"/>
        <Add mover="favournative"/>
        <Add mover="design"/>
        <Add mover="clear_all_constraints"/>
        <Add filter="ddg"/>
        <Add filter="interface_buried_sasa"/>
        <Add filter="interface_hydrophobic_sasa"/>
        <Add filter="interface_polar_sasa"/>
        <Add filter="new_buns_bb_heavy"/>        <Add filter="new_buns_sc_
heavy"/>
        <Add filter="Packstat"/>
        <Add filter="interface_Hbonds"/>
    </PROTOCOLS>
    <OUTPUT scorefxn="ref2015"/>
</ROSETTASCRIPTS>
```

## Statistical analysis

Results are expressed as means ± SEM. Statistical analysis was performed using Sigmaplot (version 14.0, Systat Software) or Igor Pro 8 (Wavemetrics). Results of in vitro experiments were analyzed using Student's t test (for differences between two groups). Results of in vivo experiments were analyzed using Two Way Repeated Measures ANOVA with Holm-Sidak post-hoc analysis. Differences between groups with $p<0.05$ were considered statistically significant. In experiments on mice technical replicates (n) were individual neurons and biological replicates (N) were individual mice. Details on statistical analysis are included in the figure legends. We calculated the sample power for rat behavioral studies with eight animals per group is needed to show significant differences of 20% or more. The acceptable power level was considered to be between 0.8 and 0.9. For the thermal hyperalgesia test we assumed the mean value for the control population is 7.5 s and we want to be able to distinguish a difference of 20% with a common standard deviation of about 10%. To test if the two populations are not equal at a significance level of 0.05, a power of 0.8 gives an n=8. The observed effect size was greater than expected and resulted in significant results with even smaller n. Investigators were blinded to identification of compound components in all studies. In brief, compound doses and vehicles were prepared and dosed on the day of the study by an independent researcher from those conducting the behavioral assessments. All treatment groups were randomized independent of baseline responses and the treatments included vehicle and positive controls were randomized on each day of assessment for blinded observers.

## Acknowledgements

We thank Dr. Neil Castle (Icagen) for providing human Na$_V$1.8 and Na$_V$1.9 stable cell lines, Dr. Stephen Waxman (Yale University) for providing rat Na$_V$1.3 expressing HEK-293 cells, Dr. Alan L Goldin (University of California, Irvine) for providing hNa$_V$1.2 cDNA, Dr. David Ginty (Harvard University) for the *Mrgprd*$^{GFP}$ *mouse line*, Dr. Daniel J Tancredi (UC Davis) for providing advice on statistical data analysis, and Drs. Scott Fishman and David J Copenhaver (UC Davis Pain Clinic) for valuable discussions of potential therapeutic applications of the designed peptides. Funding: National Institute of Neurological Disorders and Stroke HEAL Initiative Grant UG3NS114956 (VY-Y, HW, JTS, BH, KW).

# Additional information

## Competing interests

Phuong T Nguyen, Hai M Nguyen, Karen M Wagner, Bruce Hammock, Heike Wulff, Vladimir Yarov-Yarovoy: is named inventor on a patent application entitled 'Peptides targeting sodium channels to treat pain' based on this research, filed by the University of California. (U.S. provisional application no. 63/358,684, filed July 6, 2022). Anh Tuan Ton, Richard Kondo, Andre Ghetti: is affiliated with AnaBios Corporation. The author has no financial interests to declare. Michael W Pennington: is affiliated with Ambiopharm Inc. The author has no financial interests to declare. Jon T Sack: Reviewing editor, *eLife*. The other authors declare that no competing interests exist.

## Funding

| Funder | Grant reference number | Author |
|---|---|---|
| National Institute of Neurological Disorders and Stroke | UG3NS114956 | Phuong T Nguyen |

The funders had no role in study design, data collection and interpretation, or the decision to submit the work for publication.

## Author contributions

Phuong T Nguyen, Conceptualization, Data curation, Software, Formal analysis, Validation, Investigation, Visualization, Methodology, Writing – original draft, Writing – review and editing; Hai M Nguyen, Karen M Wagner, Data curation, Formal analysis, Validation, Investigation, Visualization, Methodology, Writing – original draft, Writing – review and editing; Robert G Stewart, Data curation, Validation, Investigation, Visualization, Writing – original draft, Writing – review and editing; Vikrant Singh, Mark W Lillya, Anh Tuan Ton, Data curation, Investigation, Visualization; Parashar Thapa, Data curation, Investigation, Visualization, Writing – original draft; Yi-Je Chen, Resources, Methodology; Richard Kondo, Resources, Supervision, Investigation, Methodology, Writing – original draft; Andre Ghetti, Michael W Pennington, Resources, Supervision, Methodology, Writing – original draft; Bruce Hammock, Resources, Supervision, Funding acquisition, Validation, Visualization, Methodology, Writing – original draft, Writing – review and editing; Theanne N Griffith, Formal analysis, Investigation, Visualization, Writing – review and editing; Jon T Sack, Conceptualization, Resources, Data curation, Software, Formal analysis, Supervision, Funding acquisition, Validation, Investigation, Visualization, Methodology, Writing – original draft, Writing – review and editing; Heike Wulff, Vladimir Yarov-Yarovoy, Conceptualization, Resources, Data curation, Software, Formal analysis, Supervision, Funding acquisition, Validation, Investigation, Visualization, Methodology, Writing – original draft, Project administration, Writing – review and editing

## Author ORCIDs

Phuong T Nguyen http://orcid.org/0000-0002-9461-7807
Hai M Nguyen http://orcid.org/0000-0002-1422-7041
Michael W Pennington http://orcid.org/0000-0001-5446-3447
Bruce Hammock http://orcid.org/0000-0003-1408-8317
Theanne N Griffith http://orcid.org/0000-0003-0090-6286
Jon T Sack http://orcid.org/0000-0002-6975-982X
Heike Wulff http://orcid.org/0000-0003-4437-5763
Vladimir Yarov-Yarovoy http://orcid.org/0000-0002-2325-4834

## Ethics

Research involving vertebrate animals was done at the University of California following protocols reviewed and approved by the UC Davis Institutional Animal Care and Use Committee (UCD IACUC) - Animal Welfare Assurance Number A3433-01. The animals were cared for by the Center for Laboratory Animal Science (CLAS) Veterinary Services under a currently AAALAC approved program under the direction of Dr. Laura Brignolo (Campus Veterinarian). The animals were housed in NIH-approved facilities in CLAS and are observed daily by technicians. Unusual events are reported to the on call veterinarian, as well as to the investigator according to posted protocols. Other maintenance

veterinary care was conducted according to NIH guidelines on the Use and Care of Animals. Facilities were inspected regularly according to NIH and AAALAC guidelines.

## Decision letter and Author response
Decision letter https://doi.org/10.7554/eLife.81727.sa1
Author response https://doi.org/10.7554/eLife.81727.sa2

---

## Additional files

### Supplementary files
• MDAR checklist

### Data availability
All data generated or analysed during this study are included in the manuscript.

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
