## [Editor Report]

Chronic pain is a major health issue that is notably lacking pharmacological, non-opioid treatment options. The authors of this manuscript have deployed a powerful combination of experimental and computational tools to optimize the binding and selectivity of a series of peptide toxins derived from the toxin Protoxin II. The authors succeed in obtaining a peptide with improved selectivity and affinity for the human sodium channel Nav1.7, which is a major player in pain transmission and generation. This strategy represents a major advance on the road to obtaining specific peptide inhibitors for the treatment of chronic pain.

---

## [Decision Letter]

**Decision letter after peer review:**

Thank you for submitting your article "Computational design of peptides to target Nav1.7 channel with high potency and selectivity for the treatment of pain" for consideration by *eLife*. Your article has been reviewed by 3 peer reviewers, and the evaluation has been overseen by a Reviewing Editor and Richard Aldrich as the Senior Editor. The following individuals involved in review of your submission have agreed to reveal their identity: Leon D Islas (Reviewer #1); David H Hackos (Reviewer #2).

Essential revisions:

1) The main concern relates to the measurement of the effects of the peptides on action potentials. The authors should repeat the current clamp experiments while adjusting the resting membrane voltage to -70mV to enable Nav1.7-dependent signaling.

2) The logic employed in the design of subsequent peptides is not always well explained. Please carefully respond to the criticisms of reviewer 3

*Reviewer #1 (Recommendations for the authors):*

In general the results support the conclusions presented in the Discussion section.

In lines 732-734, the statement is not supported by the data since only a very high concentration of the peptide is used and, as shown by the authors, this could be blocking other Nav channels.

It is puzzling that in mouse DRGs the effect of peptide (and TTX) is so small in the rheobase and frequency of action potentials (figure 6 supplement).

The effects of the best peptides on thermal pain and CIPN induced pain are tested by intrathecal injection of the peptides, however, one can envision a systemic injection in a clinical setting, did you try intravenous injection to asses more realistically the possible off-target effects of the peptides?

*Reviewer #2 (Recommendations for the authors):*

One potential weakness in the authors methods is in their current clamp analysis. It is interesting that 27% of these neurons could not be inhibited by PTx2-3127 or TTX. It is possible that the authors observed this effect because they didn't adjust the resting membrane voltage in their current clamp recordings. The methods section indicates that neurons were excluded if the resting membrane potential was > -40mV. But this likely means recordings were conducted from neurons with membrane voltages between -40mV and -60mV. Nav1.7 is mostly inactivated in this voltage range since Nav1.7 has a V1/2 of around -65mV. Under these conditions, it should not be surprising that neither TTX nor PTx2-3127 would have an effect on action potentials since under these conditions, APs are carried entirely by Nav1.8 channels. To properly examine block of APs by a Nav1.7 inhibitor, the experimenters should inject current to adjust the resting membrane voltage to -70mV. We have shown that this is required to see Nav1.7-dependence in AP generation (see Shields et al., J. Neurosci. 2018). While this adjustment might seem arbitrary, leaving the membrane voltage unchanged is also arbitrary since it is unclear whether the membrane voltage of the cell body following 1-2 days of cell culture bears any relationship to the membrane voltage at the fibers within the skin where stimulus detection occurs. It is our assumption that the fiber voltage is likely quite hyperpolarized given that c-fiber APs appear completely dependent on Nav1.7 in-vivo.

The authors should either mention this caveat in their results or Discussion sections or should repeat the current clamp experiments while adjusting the resting membrane voltage to -70mV to enable Nav1.7-dependent signaling. I think it is likely that a more powerful block of APs will be observed if this experiment is done correctly.

Shields SD, Deng L, Reese RM, Dourado M, Tao J, Foreman O, Chang JH, Hackos DH. Insensitivity to Pain upon Adult-Onset Deletion of Nav1.7 or Its Blockade with Selective Inhibitors. J Neurosci. 2018 Nov 21;38(47):10180-10201.

*Reviewer #3 (Recommendations for the authors):*

The manuscript entitled "Computational design of peptides to target NaV1.7 channel with high potency and selectivity for the treatment of pain" describes the molecular design of antinociceptive peptides with the aim to improve the peptide affinity towards Nav1.7, and the authors performed in vivo experimental assays of such molecular designed peptide to validate them. The authors use state-of-the-art techniques, the data presented here is clear and straight-forward and of great quality. It also covers an important topic and worthy of study so, some minor comments, and some few issues may require clarification.

Why was it speculated that the 210 PTx2-2954 peptide was misfolded? That is, an arginine vs lysine difference at position 26 is not a reason to think in a misfolded peptide.

The reasonings from ProTx-II, to PTx2-2955, to PTx2-3066 and to finally PTx2-3127 and PTx2-3258 are confusing and somewhat handpicked. For example, the use of Nor-Arg is detrimental for PTx2-2955, but finally improved in PTx2-3260. There is no doubt that the authors did a great effort synthesizing all these variants, but it is difficult to catch some ideas, and in this type of work serendipity with no regrets could be involved.

The design of PTx2-3066 is irrelevant to the narration of PTx2-2955 but to the native ProTx-II, so why not start this molecular design description from PTx2-3066 and avoid the narration of PTx2-2955. I suggest describing this work, just from ProTx-II, to PTx2-3066 and to PTx2-3127 and PTx2-3258 and mentioning that several variants were generated. The IC50 values of all variants created could be included in a table.

What was the reasoning for merging Y1Q and M19F from PTx2-3067 into PTx2-3066? PTx2-3067 had less blocking activity than PTx2-3066.

I understand that Nav1.7 is preferentially expressed in PNS, and not only in CNS; so please, discuss why the peptides were administered intrathecally and neither intravenously nor intraperitoneally. Is there any chance that the best ProTx-II variants blocking Nav1.7 could also target other ion channels such as calcium ion channels as the inhibitor ziconotide? Please discuss.

---

## [Author Response]

Reviewer #1 (Recommendations for the authors):In general the results support the conclusions presented in the Discussion section.In lines 732-734, the statement is not supported by the data since only a very high concentration of the peptide is used and, as shown by the authors, this could be blocking other Nav channels.

Good point. We have changed the sentence to read "endogenous Na_V_ channels” and added shortly thereafter: "We note that 1 µM PTx2-3127 partially inhibits hNa_V_1.6 in HEK cells, and other hNa_V_s to a lesser extent (Table 1), raising the possibility of off-target Navs contributing to neuronal modulation by 1 µM PTx2-3127."

It is puzzling that in mouse DRGs the effect of peptide (and TTX) is so small in the rheobase and frequency of action potentials (figure 6 supplement).

We were surprised by the insensitivity as well, yet it is persistent in this preparation of MrgprD^GFP^ neurons. We conducted a new set of current clamp experiments while injecting current to hold the membrane voltage more negative to disinactivate Nav1.7 and (new Figure 6 Supplement 2) to test whether more peptide and TTX sensitivity might be revealed. However, we saw a similar result, and now report: "Injecting current into DRG neurons to lower resting potential can relieve Nav1.7 inactivation and enhance the reliance of action potential generation on Nav1.7 conductance (REF: https://pubmed.ncbi.nlm.nih.gov/30301756/ ). However even when currents were injected to hold MrgprD^GFP^ neurons at more negative than -80 mV (after liquid junction potential correction), we saw similar results with PTx2-3127 (Figure 6—figure supplement 2), and in 25% of neurons (4 of 16), no block of action potentials was observed in TTX."

The effects of the best peptides on thermal pain and CIPN induced pain are tested by intrathecal injection of the peptides, however, one can envision a systemic injection in a clinical setting, did you try intravenous injection to asses more realistically the possible off-target effects of the peptides?

Previously, a lack of efficacy of systemically administered ProTx-II was explained with insufficient penetration of the perineurial barrier.

(Schmalhofer WA, et al. ProToxin-II, a selective inhibitor of NaV1.7 sodium channels, blocks action potential propagation in nociceptors. Mol Pharmacol. 2008;74:1476–1484). We did test a direct and local subcutaneous injection of ProTx-II against a carrageenan model of inflammatory pain which demonstrated the benefit of targeting the nervous system directly by intrathecal injection of ProTx-II and the designed peptides.

Reviewer #2 (Recommendations for the authors):One potential weakness in the authors methods is in their current clamp analysis. It is interesting that 27% of these neurons could not be inhibited by PTx2-3127 or TTX. It is possible that the authors observed this effect because they didn't adjust the resting membrane voltage in their current clamp recordings. The methods section indicates that neurons were excluded if the resting membrane potential was > -40mV. But this likely means recordings were conducted from neurons with membrane voltages between -40mV and -60mV. Nav1.7 is mostly inactivated in this voltage range since Nav1.7 has a V1/2 of around -65mV. Under these conditions, it should not be surprising that neither TTX nor PTx2-3127 would have an effect on action potentials since under these conditions, APs are carried entirely by Nav1.8 channels. To properly examine block of APs by a Nav1.7 inhibitor, the experimenters should inject current to adjust the resting membrane voltage to -70mV. We have shown that this is required to see Nav1.7-dependence in AP generation (see Shields et al., J. Neurosci. 2018). While this adjustment might seem arbitrary, leaving the membrane voltage unchanged is also arbitrary since it is unclear whether the membrane voltage of the cell body following 1-2 days of cell culture bears any relationship to the membrane voltage at the fibers within the skin where stimulus detection occurs. It is our assumption that the fiber voltage is likely quite hyperpolarized given that c-fiber APs appear completely dependent on Nav1.7 in-vivo.The authors should either mention this caveat in their results or Discussion sections or should repeat the current clamp experiments while adjusting the resting membrane voltage to -70mV to enable Nav1.7-dependent signaling. I think it is likely that a more powerful block of APs will be observed if this experiment is done correctly.Shields SD, Deng L, Reese RM, Dourado M, Tao J, Foreman O, Chang JH, Hackos DH. Insensitivity to Pain upon Adult-Onset Deletion of Nav1.7 or Its Blockade with Selective Inhibitors. J Neurosci. 2018 Nov 21;38(47):10180-10201.

Thank you for sharing your experience and such a thoughtful suggestion. As suggested, we conducted a new set of current clamp experiments while injecting current to hold the membrane voltage more negative to disinactivate Nav1.7 and to test whether more peptide and TTX sensitivity might be revealed. However, we saw a similar result, and now report: "Injecting current into DRG neurons to lower resting potential can relieve Nav1.7 inactivation and enhance the reliance of action potential generation on Nav1.7 conductance (REF: https://pubmed.ncbi.nlm.nih.gov/30301756/). However even when currents were injected to hold MrgprD^GFP^ neurons at more negative than -80 mV (after liquid junction potential correction), we saw similar results with PTx2-3127 (Figure 6—figure supplement 2), and in 25% of neurons (4 of 16), no block of action potentials was observed in TTX."

Reviewer #3 (Recommendations for the authors):The manuscript entitled "Computational design of peptides to target NaV1.7 channel with high potency and selectivity for the treatment of pain" describes the molecular design of antinociceptive peptides with the aim to improve the peptide affinity towards Nav1.7, and the authors performed in vivo experimental assays of such molecular designed peptide to validate them. The authors use state-of-the-art techniques, the data presented here is clear and straight-forward and of great quality. It also covers an important topic and worthy of study so, some minor comments, and some few issues may require clarification.1. Why was it speculated that the 210 PTx2-2954 peptide was misfolded? That is, an arginine vs lysine difference at position 26 is not a reason to think in a misfolded peptide.

Our collaborators at AmbioPharm who synthesized and folded all of our ProTx-II variants (Michael Pennington is a co-author on our manuscript and Chief Scientific Officer at AmbioPharm) reported that PTx2-2954 peptide synthesis product had multiple peaks and the isolated peak most likely was a misfolded product due to its lack of inhibition of hNav1.7. PTx2-2955 peptide synthesis product had the typical ProTx-II folding pattern and inhibited hNav1.7. Notably, we isolated and tested only one dominant peptide synthesis product for each ProTx-II variant. We decided to remove the speculative sentence and replaced it with the following sentence:

“We currently have no explanation for why the PTx2-2954 peptide was not active on hNaV1.7 despite having only an arginine versus lysine difference at position 26.”

2. The reasonings from ProTx-II, to PTx2-2955, to PTx2-3066 and to finally PTx2-3127 and PTx2-3258 are confusing and somewhat handpicked. For example, the use of Nor-Arg is detrimental for PTx2-2955, but finally improved in PTx2-3260. There is no doubt that the authors did a great effort synthesizing all these variants, but it is difficult to catch some ideas, and in this type of work serendipity with no regrets could be involved.

Please note that the reasoning for our narration from ProTx-II, to PTx2-2955, to PTx2-3066, to PTx2-3127, and to PTx2-3258 is based on learning in each optimization round which particular combination of mutations results in the most potent and selective ProTx-II redesign. Specifically:

1. PTx2-2955 included V20R, K26R, and K28E mutations compared to the wild-type ProTx-II which ultimately benefited the potency and selectivity of PTx2-3127 and PTx2-3258. Mutations W5A, M6F, and R22norR did not result in more potent and selective peptide and were eliminated in the following rounds of optimization.

2. PTx2-3066 included W7Q, S11K, E12D, and W30L mutations compared to PTx2-2955 which ultimately benefited the potency and selectivity of PTx2-3127 and PTx2-3258. Mutation M19L did not improve potency and selectivity and was eliminated in the following rounds of optimization.

3. PTx2-3127 included the M19F mutation compared to PTx2-3066 which ultimately benefited the potency and selectivity of PTx2-3258. Mutation Y1Q did not improve potency and selectivity and was eliminated in the following round of optimization.

4. PTx2-3258 included the Y1H mutation compared to PTx2-3127 which ultimately and selectivity.

We think that including beneficial and nonbeneficial mutations from all four rounds of optimization is essential for a rigorous and reproducible representation of the data. Notably, some of the specific tested mutations (Y1Q, W5A, M6F, M19L, and R22norR) have not made it to the final optimization round due to having lower potency and/or selectivity compared to the top ProTx-II redesigns in each optimization round. We now added related text into each optimization round and Table 5 to summarize key mutations introduced and eliminated in the top ProTx-II redesigns after each optimization round:

3. The design of PTx2-3066 is irrelevant to the narration of PTx2-2955 but to the native ProTx-II, so why not start this molecular design description from PTx2-3066 and avoid the narration of PTx2-2955. I suggest describing this work, just from ProTx-II, to PTx2-3066 and to PTx2-3127 and PTx2-3258 and mentioning that several variants were generated. The IC50 values of all variants created could be included in a table.

Please see our response to Comment 2 above. Thank you for your suggestion – we added Table 1 to include IC50 values for all ProTx-II peptide variants presented in the manuscript.

4. What was the reasoning for merging Y1Q and M19F from PTx2-3067 into PTx2-3066? PTx2-3067 had less blocking activity than PTx2-3066.

We explored incorporating Y1Q and M19F into PTx2-3066 in our 3^rd^ optimization round by creating the PTx2-3127 peptide to test if these mutations would further improve the potency and selectivity of PTx2-3066. Notably, PTx2-3127 has better potency and selectivity compared to PTx2-3066.

5. I understand that Nav1.7 is preferentially expressed in PNS, and not only in CNS; so please, discuss why the peptides were administered intrathecally and neither intravenously nor intraperitoneally. Is there any chance that the best ProTx-II variants blocking Nav1.7 could also target other ion channels such as calcium ion channels as the inhibitor ziconotide? Please discuss.

Although the PNS is more accessible to the blood stream than the CNS, under homeostatic circumstances there is still a blood-nerve barrier of the perineurium around peripheral nerves. Previously, a lack of efficacy of systemically administered ProTx-II was explained with insufficient penetration of the perineurial barrier.

(Schmalhofer WA, et al. ProToxin-II, a selective inhibitor of NaV1.7 sodium channels, blocks action potential propagation in nociceptors. Mol Pharmacol. 2008;74:1476–1484.)

With intravenous injection in rodents, there is a hazard of rapid delivery to the heart and organs before the peptides could reach the target neurons. The risk of toxic side effects from rapid delivery could complicate behavioral endpoints that are meant to reflect the action at the targeted nociceptors. Intraperitoneal injections also often have poorly described PK in terms of the mechanisms whereby agents get systemic exposure after ip. injection. There is again the complication that the systemic exposure that does occur may insufficiently penetrate the perineurial barrier.

To ensure the in vivo efficacy of the synthesized peptides in our initial studies we choose to ensure the peptides reached the targeted Nav1.7 channels on DRG neurons via intrathecal injection.

There is indeed a possibility that our lead peptides might inhibit other ion channels, but with relatively low affinity. For example, our lead peptides (PTx2-3127 and PTx2-3258) inhibited hERG K^+^ channel with IC50 around 1.9 µM (see Table 2). We added the following related sentence at the end of the “4^th^ optimization round” section: “Notably, while the wild-type ProTx-II did not inhibit K_V_2.1 channel at 100 nM (Bosmans, Martin-Eauclaire, and Swartz, 2008; Bosmans, Puopolo, Martin-Eauclaire, Bean, and Swartz, 2011; Schmalhofer et al., 2009), it inhibited Cav3 channels in the μM range (Bladen, Hamid, Souza, and Zamponi, 2014; Middleton et al., 2002). We hypothesize that our lead peptides (PTx2-3127 and PTx2-3258) might also inhibit Cav3 channels in the μM range and further optimization of peptide selectivity and potency will be needed.” Notably, ziconotide has a very different structural fold compared to ProTx-II and binds to a different structural region – the pore-forming domain of Cav2.2 channel (https://pubmed.ncbi.nlm.nih.gov/34234349/ ) compared to the VSD-II and VSD-IV on hNav1.7 (https://pubmed.ncbi.nlm.nih.gov/35878056/ ).